

Manuscript prepared for Geosci. Model Dev.

Date: 6 June 2018

# PCR-GLOBWB 2: a 5 arc-minute global hydrological and water resources model

Edwin H. Sutanudjaja[1], Rens van Beek[1], Niko Wanders[1], Yoshihide Wada[1,2], Joyce H.C. Bosmans[1], Niels Drost[3], Ruud J. van der Ent[1], Inge E. M. de Graaf[4], Jannis M. Hoch[1,5], Kor de Jong[1], Derek Karssenberg[1], Patricia López López[1,5], Stefanie Peßenteiner[6], Oliver Schmitz[1], Menno W. Straatsma[1], Ekkamol Vannametee[7], Dominik Wisser[8,9], and Marc F. P. Bierkens[1,10]

Department of Physical Geography, Faculty of Geosciences, Utrecht University, Utrecht, The Netherlands
International Institute for Applied Systems Analysis, Laxenburg, Austria
Netherlands eScience Center, Amsterdam, The Netherlands
Chair of Environmental Hydrological Systems, Faculty of Environment and Natural Resources, University of Freiburg, Freiburg, Germany
Unit Inland Water Systems, Deltares, Delft, The Netherlands
Department of Geography and Regional Science, University of Graz, Graz, Austria
Department of Geography, Chulalongkorn University, Bangkok, Thailand
Food and Agriculture Organization of the United Nations, Rome, Italy
Institute for the Study of Earth, Oceans, and Space, University of New Hampshire, New Hampshire, USA
Unit Soil and Groundwater Systems, Deltares, Utrecht, The Netherlands

Correspondence to: E. H. Sutanudjaja (E.H.Sutanudjaja@uu.nl)

## Abstract

We present PCR-GLOBWB 2, a global hydrology and water resources model. Compared to previous versions of
PCR-GLOBWB, this version fully integrates water use. Sector-specific water demand, groundwater and surface
water withdrawal, water consumption and return flows are dynamically calculated at every time step and interact
directly with the simulated hydrology. PCR-GLOBWB 2 has been fully rewritten in Python and PCRaster-Python
and has a modular structure, allowing easier replacement, maintenance, and development of model components.
PCR-GLOBWB 2 has been implemented at 5 arc-minute resolution, but a version parameterized at 30 arc-minute
resolution is also available. Both versions are available as open source codes on https://github.com/UU-
Hydro/PCR-GLOBWB_model. PCR-GLOBWB 2 has its own routines for groundwater dynamics and surface
water routing.  These relatively simple routines can alternatively be replaced by dynamically coupling PCR-
GLOBWB 2 to a global two-layer groundwater model and 1D-2D-hydrodynamic models, respectively. Here, we
describe the main components of the model, compare results of the 30 arc-minute and the 5 arc-minute versions
and evaluate their model performance using GRDC discharge data. Results show that model performance of the 5
arc-minute version is notably better than that of the 30 arc-minute version. Furthermore, we compare simulated
time series of total water storage (TWS) of the 5 arc-minute model with those observed with GRACE, showing
similar negative trends in areas of prevalent groundwater depletion. Also, we find that simulated total water
withdrawal, matches reasonably well with reported water withdrawal from AQUASTAT, while water withdrawal
by source and sector provide mixed results.

# 1 Introduction

The last decades saw the development of an increasing number of global hydrological models (GHMs), e.g. VIC (Liang et al., 1994, Nijssen et al., 2001), WMB (Fekete et al., 2002), WaterGAP (Döll et al., 2003), H08 (Hanasaki et al., 2008a, Hanasaki et al., 2018), MAC-PDM (Gosling and Arnell, 2011) (see Bierkens et al., 2014, Bierkens, 2015 and Kauffeldt et al. 2016 for a more extensive list, also including land surface models). GHMs have become essential tools to quantify and understand the global terrestrial water cycle, as they simulate the distributed hydrological response to weather and climate variations at higher resolution (typically $0.5^o \times 0.5^o$) than used previously in general circulation models (GCMs), with more sophisticated runoff generation processes and river routing.  As such, global hydrological models have been used for medium-range to seasonal flood forecasting (Bierkens and van Beek, 2009, Alfieri et al., 2013, Candogan Yossef et al., 2013) as well as for a myriad of water-related global change assessments. Examples are: the projection or estimation of future flood and drought events (Sperna-Weiland et al., 2012, Dankers et al., 2013, Prudhomme et al., 2013, Wanders et al. 2015, Wanders and Wada, 2016), current and future flood hazard and risk (Pappenberger et al., 2012, Hirabayashi et al., 2013, Ward et al., 2013, Winsemius et al., 2013, 2016), global groundwater depletion (Wada et al., 2010, Gleeson et al., 2012), the contribution of terrestrial water stores to global sea level change (Konikow, 2011, Wada et al., 2012, Pohkrel et al., 2013), current and future water scarcity under climate change and increasing population growth (Hanasaki et al., 2008b, Wada et al., 2011a, 2011b, Schewe et al., 2014, Haddeland et al., 2014, Wada and Bierkens, 2014), tele-connections between climate oscillations and water availability (Wanders and Wada, 2015), the impact of land use change on global water resources (Rost et al., 2008, Sterling et al., 2015, Bosmans et al., 2017) and trends in surface water temperature and cooling water potential (van Beek et al., 2012, van Vliet et al., 2012). More recently, the output from global hydrological models has been extended to study socioeconomic impacts, such as virtual water trade (Konar et al., 2013, Dalin et al., 2017) and future agricultural production (Elliott et al., 2013). These applications show that GHMs have become invaluable tools in support of global change research and environmental assessments.

PCR-GLOBWB (PCRaster GLOBal Water Balance) (van Beek and Bierkens, 2009, van Beek et al. 2011) is one of the recently developed GHMs. PCR-GLOBWB is a grid-based global hydrological model developed at the Department of Physical Geography, Faculty of Geosciences, Utrecht University, the Netherlands. The model, describing the terrestrial part of the hydrological cycle, was first introduced in a technical report by van Beek and Bierkens (2009) and then formally published in a paper of van Beek et al. (2011), focusing on global water availability issues. PCR-GLOBWB was originally developed to solve the global daily surface water balance with a spatial resolution of 30 arc-minutes (about 50 km by 50 km at the equator) and compare the resulting fresh water availability with monthly sectoral water demand in order to assess global-scale water scarcity (van Beek et al., 2011, Wada et al., 2011a,b). In this first version of PCR-GLOBWB (called PCR-GLOBWB 1 hereafter), similar to other global-scale hydrological models, water demand and water availability are treated independently, i.e. without direct feedback between human water use and other terrestrial water fluxes (e.g. Döll and Siebert, 2002, Wisser et al., 2010). Since it was first introduced, PCR-GLOBWB has been applied extensively in global water resources assessment studies. For instance, a recent search on Scopus (accessed on 13 April 2018) on the key-word "PCR-GLOBWB" yielded 113 publications with collectively over 2500 citations. Since the first version, several new

model features have been introduced such as a comprehensive water demand and irrigation module (Wada et al.,
2011b, 2014), a scheme for dynamic allocation of sectoral water demand to available surface water and
groundwater resources and the associated calculation of return flow (de Graaf et al., 2014). These features
essentially introduced a two-way interaction between water demand, water withdrawal, water consumption and
availability, particularly over irrigated areas where water demand is large and return flow is significant.
Nevertheless, all of these preceding studies using PCR-GLOBWB were performed at a relatively coarse resolution
of 30 arc-minutes, limiting their sub-regional or local applications. Additionally, some added functionalities, such
as the possibility to couple the land surface component of PCR-GLOBWB to a global MODFLOW-based
groundwater model (Sutanudjaja et al., 2011, 2014, de Graaf et al., 2015, 2017) and an extension to simulate
surface water temperature (van Beek et al., 2012), were incorporated in different versions based on the original
PCR-GLOWB 1, leading to divergent model code development.

The objective of this paper is to summarize and present the new version of the model, PCR-GLOBWB 2, which
consolidates all components that have been developed since the original version of the model was first introduced
(van Beek et al., 2011). The new version of the model, PCR-GLOBWB 2, which is able to simulate the water
balance at a finer spatial resolution of 5 arc-minutes, supersedes the original PCR-GLOBWB 1, which has a
resolution of 30 arc-minutes only[1]. The finer resolution of PCR-GLOBWB 2 allows a much better representation
of the effects of spatial heterogeneity in topography, soils, and vegetation on terrestrial hydrological dynamics
(Wood et al., 2011, Bierkens et al., 2014). Likewise, it provides a better resolution for visualization that allows
stakeholders and decision makers to assess model simulation output more easily and directly for the places they are
specifically interested in (Sheffield et al., 2010, Beven and Cloke, 2012). To assess the possible improvements, this
paper also presents the first evaluation results from the simulation of PCR-GLOBWB 2 at 5 arc-minute resolution
and compares them to a 30 arc-minute version. As discharge data are commonly used in hydrological model
performance evaluation, the simulated river discharge of PCR-GLOBWB 2 is compared to in situ discharge
observations from the Global Runoff Data Centre (GRDC, 2014).

The paper is organized as follows. Section 2 provides a global description of PCR-GLOBWB 2, including its
model structure and the new components and functionalities that have been added since PCR-GLOBWB 1. In
section 3 the global application of PCR-GLOBWB 2 is demonstrated and the results from a 58-year simulation
(1958-2015) are evaluated against observations of discharge, total water storage and reported withdrawal data.
Section 4 summarizes and concludes this paper and discusses possible future developments. Section 5 provides
information about availability of the model code and the underlying data.


---

[1] Note that Wada et al. (2016) made a preliminary version of the model that operates at 6 arc-minutes.

## 2. PCR-GLOBWB 2 – Model description

### 2.1 General overview

PCR-GLOBWB 2 is a state-of-the-art grid-based global hydrology and water resources model. It is a component-based model implementation in Python using open source PCRaster Python routines (Karssenberg et al., 2010, http://pcraster.geo.uu.nl/). The code is distributed through Github. The computational grid covers all continents except Greenland and Antarctica. Currently two versions are available: one with a spatial resolution of 5 arc-minutes in latitude and longitude and one with a coarser resolution of 30 arc-minutes. Typical time steps for hydrology and water use are one-day while sub-daily time stepping is used for hydrodynamic river routing. For all dynamic processes involved, PCR-GLOWB 2 uses a time-explicit scheme.  For each grid cell and each time step, PCR-GLOBWB 2 simulates moisture storage in two vertically stacked upper soil layers ($S_1+S_2$ in Figure 1), as well as the water exchange between the soil, the atmosphere and the underlying groundwater reservoir ($S_3$ in Figure 1). The exchange with the atmosphere comprises of precipitation, evaporation from soils, open water, snow and soils and plant transpiration, while the model also simulates snow accumulation and snowmelt. Sub-grid variability of land use, soils and topography is included and influences the schemes for runoff-infiltration partitioning, interflow, groundwater recharge (from $S_2$ to $S_3$) and capillary rise (from $S_3$ to $S_2$). Runoff, generated by snowmelt, surface runoff, interflow and baseflow, is routed across the river network to the ocean or endorheic lakes and wetlands. Routing can either be simple accumulation, simplified dynamic routing using a method of characteristics, or kinematic wave routing. In case the kinematic wave routing is used, it is also possible to use a (simplified) floodplain inundation scheme and to simulate the surface water temperature.

PCR-GLOBWB 2 includes a simple reservoir operation scheme that is applied to over roughly 6000 manmade reservoirs from the GranD database (Lehner et al., 2011), which are progressively introduced according to their construction year. Human water use is fully integrated within the hydrological model, meaning that at each time step: 1) water demands are estimated for irrigation, livestock, industry and households, 2) these demands are translated into actual withdrawals from groundwater, surface water (rivers, lakes and reservoirs) and desalinization, subject to availability of these resources and maximum groundwater pumping capacity in place, 3) consumptive water use and return flows are calculated per sector.

As an option PCR-GLOBWB 2 can be partially or fully coupled to a two-layer global groundwater model based on MODFLOW (de Graaf et al, 2017). Recent work (Hoch et al., 2017a,b) also includes coupling PCR-GLOBWB 2 to either Delft3D Flexible Mesh (Kernkamp et al., 2011) or LISFLOOD-FP (Bates et al., 2010) which are model codes that can be used to solve the 1D-2D shallow water equations  (or approximations thereof) for detailed inundation studies.

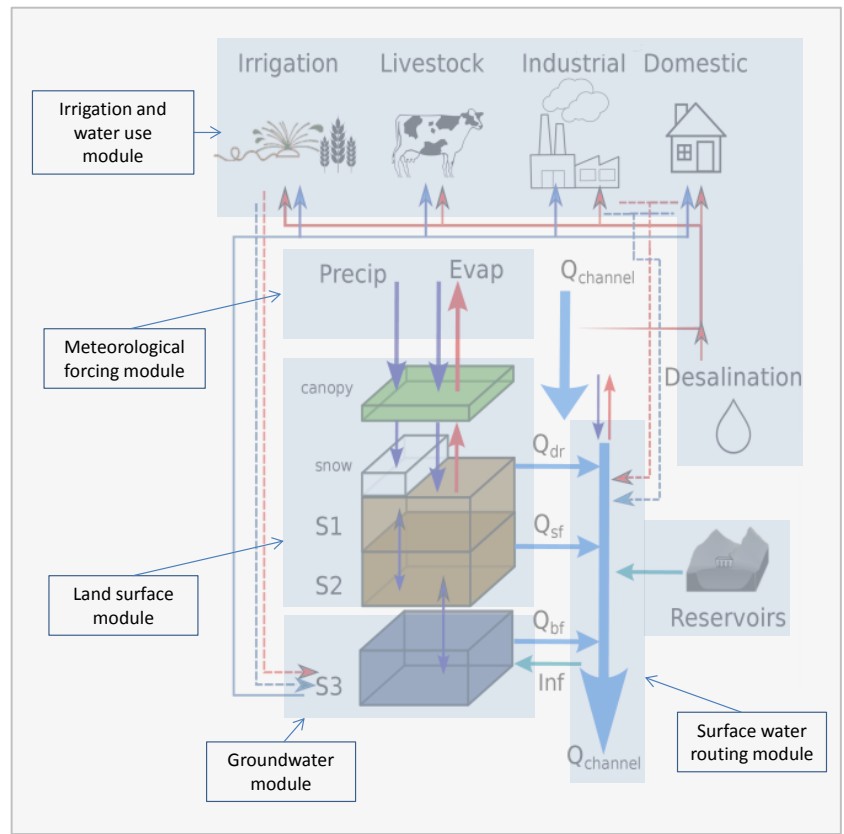

164

*Figure 1. Schematic overview of a PCR-GLOBWB 2 cell and its modelled states and fluxes. $S_1$, $S_2$ (soil moisture*

*storage), $S_3$ (groundwater storage), $Q_{dr}$ (surface runoff – from rainfall and snowmelt), $Q_{sf}$ (interflow or stormflow),*

*$Q_{bf}$ (baseflow or groundwater discharge), Inf (riverbed infiltration from to groundwater). The thin red lines*

*indicate surface water withdrawal, the thin blue lines groundwater abstraction, the thin red dashed lines return*

*flows from surface water use and the thin dashed blue lines return flows from groundwater use surface. For each*

*sector: withdrawal - return flow = consumption. Water consumption adds to total evaporation. In the figure, the*

*five modules that make up PCR-GLOBWB 2 is portrayed on the model components.*



**2.2 Model structure and flexibility**

PCR-GLOBWB 2 has a flexible modular structure (Figure 1). The modular structure of PCR-GLOBWB 2, both in
terms of model concepts and implementation (separate modules are called from a main program), makes it easy to
modify or replace components according to specific objectives of the model application, to introduce new modules
or components within the modelling system and to couple it to existing codes.

There are currently five main hydrological modules in PCR-GLOBWB 2 as illustrated in Figure 1 and briefly
described in Section 2.3: Meteorological forcing, Land surface, Groundwater, Surface water routing, Irrigation and
water use. For an extensive description of the underlying equations and methods used in each of these modules we
refer to the following sources:

• Meteorological forcing module: van Beek (2008, http://vanbeek.geo.uu.nl/suppinfo/vanbeek2008.pdf)
• Land surface module, groundwater module and surface water routing module: van Beek and Bierkens (2009,
http://vanbeek.geo.uu.nl/suppinfo/vanbeekbierkens2009.pdf), van Beek et al. (2011,
http://dx.doi.org/10.1029/2010WR009791)
• Irrigation and water use module:
o Calculation of water demand: Wada et al., (2014, https://doi.org/10.5194/esd-5-15-2014)
o Calculation of water withdrawal, consumption and return flows: de Graaf et al. (2014,
https://doi.org/10.1016/j.advwatres.2013.12.002), Wada et al. (2014, https://doi.org/10.5194/esd-5-
15-2014), Erkens and Sutanudjaja (2015, https://doi.org/10.5194/piahs-372-83-2015)

Furthermore: for details about coupling to MOFLOW we refer to:
• One-way coupling: Sutanudjaja et al. (2011, https://doi.org/10.5194/hess-15-2913-2011), de
Graaf et al. (2017, https://doi.org/10.1016/j.advwatres.2017.01.011)
• Two-way coupling: Sutanudjaja et al. (2014, http://dx.doi.org/10.1002/2013WR013807)
**2.3 Description of the modules**

Hereafter, we briefly describe the main features of the five modules. Additionally, a (non-exhaustive) list of the
model state and flux variables is provided in Table A1, whereas Table A2 lists the model inputs and parameters,
including their sources.

 **2.3.1 Meteorological forcing module**


Meteorological forcing of PCR-GLOBWB 2 uses time series of spatial fields of precipitation, temperature and
reference evaporation. Reference potential evaporation can be prescribed or calculated within the model, and is
used in the land surface module to calculate land-cover specific potential evaporation based on crop factors of the
various land cover types according to the FAO guidelines (Allen et al., 1998). There are two options for
calculating reference potential evaporation: 1) using Hamon (1963) in case only daily mean temperature is
available, 2) using Penman-Monteith following the FAO guidelines (Allen et al., 1998) if net radiation, wind speed
and vapour pressure deficit are additionally available. See van Beek et al. (2008) for details. The resulting land-
cover specific potential evaporation is subsequently used to compute the actual evaporation for different land cover
types in each cell. Apart from the calculation of evaporation, temperature is also used to partition precipitation into
snow and rain and to drive snowmelt.


**2.3.2. Land surface module**

This core module of PCR-GLOBWB 2 covers the land-atmosphere exchange, the vertical flow between soil
compartments and the eventual groundwater recharge, snow and interception storage and the runoff generation
mechanisms. These processes are simulated over a number of land cover types and aggregated proportionally
based on land cover fractions within a model cell. Users can specify their own land cover classification and
introduce their own land cover parameterization. The number of land cover types is configurable. The standard
parameterization of PCR-GLOBWB 2 carries four land cover types consisting of tall natural vegetation, short
natural vegetation, non-paddy irrigated crops, and paddy irrigated crops (i.e. wet rice). There is also a
parameterization set for six land cover types (Bosmans et al., 2017), albeit still at 30 arc minute resolution only,
that includes distinct types for pasture and rain-fed crops. For the standard four land cover parameterization of
PCR-GLOBWB, applied in this paper, the land cover types of pasture and rain-fed crops are integrated into the
short natural vegetation type.

For each land cover type, separate soil conditions can be specified. It should be noted that the soil and vegetation
conditions are in any case fully spatially distributed. Thus, vegetation properties (e.g., crop factor, Leaf Area
Index) and soil properties (depth, saturated hydraulic conductivity, etc.) vary not only between land cover types,
but may also vary from cell-to-cell (e.g., per climate zone). In the standard parameterization vegetation properties
vary over the year using a monthly climatology of phenology and crop calendars (i.e. for the crop factor and LAI).
The application of irrigation water for paddy and non-paddy irrigation is done by the irrigation and water use
module. It is based on the FAO guidelines of Allen et al. (1998) and is dependent on the actual soil water storage
($S_1$, $S_2$) or paddy-open water storages. All fluxes, from and to the land surface module in Figure 1, are thus
calculated separately per land cover type. The resulting vertical fluxes for each land cover type are: interception
evaporation, bare soil evaporation, snow sublimation, vegetation-specific transpiration. In the soil column, vertical
fluxes are driven by degrees of saturation of soil layers and interact with the underlying groundwater store, $S_3$ (see
e.g. van Beek and Bierkens, 2009; Sutanudjaja et al., 2011; Sutanudjaja 2012 for detailed explanation). Surface

runoff ($Q_{dr}$, from precipitation and snowmelt) consists of infiltration excess runoff and saturation excess runoff following a sub-grid approach that mimics variable source areas, i.e. the improved Arno Scheme (Todini, 1996, Hagemann and Gates, 2003). Interflow or stormflow ($Q_{sf}$), mostly occurring in regolith soils on hillslopes, is also handled with a sub-grid approach based on a runoff parameterization by Sloan and Moore (1984). All fluxes are computed per land cover type and balanced with the available storage to arrive at the net flux that is used to update the storages for the next time step. Also, to report the overall fluxes per cell, and to pass these to other modules, the land cover specific fluxes are subsequently averaged (weighted by land cover type fractions).

For the standard parameterization of the land surface module the following data sets are combined (see Table A2): the cell fractions of various non-irrigation land cover types are based on the map of Global Land Cover Characteristics Data (GLCC) Base Version 2.0 (Loveland et al., 2000) with the land cover classification following Olson (1994a, b) and the parameter sets from Hagemann et al. (1999) and Hagemann (2002). Irrigation land cover types (i.e. paddy and non-paddy), including their crop calendars and growing season lengths, are parameterized based on the data set of MIRCA2000 (Portmann et al., 2010) and the Global Crop Water Model of Siebert and Döll (2010). We refer to van Beek et al. (2011) for detailed descriptions.

**2.3.3. Groundwater module**

The groundwater module calculates groundwater storage dynamics subject to recharge and capillary rise (calculated by the land surface module), groundwater discharge ($Q_{bf}$, in case of a positive groundwater storage) and riverbed infiltration (Inf). Groundwater discharge (assumed the same as groundwater baseflow here) depends on a linear storage-outflow relationship ($Q_{bf} = S_3/J$) where the proportionality constant $J$ is calculated following drainage theory of Kraijenhoff-van de Leur (1958) based on drainage network density and aquifer properties. Riverbed infiltration occurs only in case $Q_{bf}$ becomes 0 by groundwater withdrawal. Under persistent groundwater withdrawal (calculated with the Irrigation and Water use module) that is larger than the sum of recharge and riverbed infiltration, the groundwater storage $S_3$ is allowed to become negative. In this case, the part of the withdrawn groundwater in excess of the input (recharge and riverbed infiltration) is seen as non-renewable groundwater withdrawal leading to groundwater depletion (permanent loss of groundwater from storage). In case withdrawal becomes smaller than the input, the remaining input is used to first fill the negative storage to zero, before baseflow $Q_{bf}$ commences again. As an alternative, it is also possible to limit the maximum volume of non-renewable groundwater that can be extracted. .

It is possible to use a full-fledged groundwater flow model based on MODFLOW (Harbaugh et al., 2000) coupled to PCR-GLOBWB 2 in order to calculate groundwater heads and flow paths. This can be done as a one-way coupling where PCR-GLOWB 2 is first run with the standard groundwater module (reservoir $S_3$ with only vertical fluxes) to yield time series of net groundwater recharge (recharge – capillary rise) and surface water levels. These fluxes/inputs are subsequently used to force the groundwater flow model (see e.g. Sutanudjaja et al., 2011, de Graaf et al., 2017). Another possibility is to use a two-way coupling where the groundwater module of PCR-GLOBWB 2 is replaced by the groundwater flow model. In this case, at each time

step fluxes are exchanged between the groundwater model and the land surface module, and the groundwater
model and the surface water routing module (Sutanudjaja et al. 2014).


**2.3.4 Surface water routing module**

Following an 8-point steepest gradient algorithm across the terrain surface (local drainage direction or LDD), all
cells of the modelled domain are connected to a strictly convergent drainage network that together makes up the
river basins and sub-basins of the model domain. The lowermost cell is either connected to the ocean or to an
endorheic basin. Per cell, the sum of the three daily runoff fluxes (Figure 1) is aggregated and routed along the
drainage network until passing the lowermost cell and being removed from the model. Routing can be done in
three ways of increasing complexity: 1) simple accumulation of the fluxes over the drainage network; 2) a travel-
time characteristic solution (Karssenberg et al., 2007), and 3) the kinematic wave solution.

The first method is typically aggregated over longer time steps (e.g. month or year) that are larger than the travel
times of water along the longest river length. The second routing method includes an estimation of cell flow
velocity based on average discharge from the last 5 years and Manning's equation, which assumes the energy slope
to be equal to the bed slope. This estimated velocity is used to move the volume of water in the channel of a cell
the corresponding distance within one daily time step along the drainage network. This method works reasonably
well for relatively steep rivers in humid climates where the friction slope is close to the bed slope and the rivers are
equally filled with water throughout the year. The third method is the kinematic wave approximation of the Saint
Venant equations with flow described by Manning's equation, Also, here, it is assumed that friction slope and bed
slope are equal, which makes it valid for rivers without backwater effects. The kinematic wave is solved using a
time-explicit variable sub-time stepping scheme based on the minimum Courant number. Of these methods, the
kinematic wave solution simulates the propagation of the flood wave more realistically while the others provide an
expedient means to approximate discharge over longer periods.

Using the kinematic wave method, it is possible to model floodplain inundation which occurs if the discharge
exceeds the bankfull capacity of a channel. The excess discharge volume is spread over the entire cell from the
lowest part of the cell (based on a higher resolution sub-grid DEM) yielding a flooded area with an approximated
flood depth. In case of flooding, the simulated river flow is impacted by adjusting the wetted area and wetted
perimeter and calculating a weighted Manning coefficient from the individual Manning coefficients of the
floodplains and the channel.

Lakes and reservoirs are part of the drainage network. Lakes and reservoirs can extend over multiple cells, in
which case the storage is subdivided by area such as to ensure that lake and reservoir levels are the same across
their extent. The active storage of lakes and the actual storage of reservoirs are dynamically updated, for the lake
outflow a standard storage-outflow relationship based on a rectangular cross-section over a broad-crested weir
(Bos, 1989) is used, while reservoirs follow a release strategy. This strategy is, by default, aimed at passing the
average discharge, while maintaining levels between a minimum and maximum storage (Wada et al., 2014), but
more elaborate strategies that take account of downstream water demand are possible (e.g. van Beek et al., 2011).
Lakes and reservoir areas change based on global volume-area relationships. All surface water areas, which can be
classified into several water types, river channels, inundated floodplains, lakes and reservoirs, are subject to open
water evaporation calculated from reference potential evaporation multiplied with factors depending on water
types and depths. Moreover, surface waters are subject to surface water withdrawal calculated with the Irrigation
and Water Use module.

If the kinematic wave approach is used, it can be also augmented with an energy routing scheme to simulate
surface water temperature (van Beek et al., 2012). Finally, it should be noted that it is possible to run the routing
routine from PCR-GLOBWB 2 as a stand-alone routine, which allows it to be fed with the specific discharge from
other land surface models.

The routing methods that are available in PCR-GLOBWB 2 will yield significant errors for wide lowland rivers
where backwater effects are important. In this case, it is possible to replace the surface water module for part of the
modelling domain with hydrodynamic models solving the shallow water equations (Hoch et al., 2017a). Hoch et al.
(2017b) developed a generic coupler for this purpose that enables coupling to multiple hydrodynamic modelling
codes (https://doi.org/10.5281/zenodo.597107).

Although any data set can be used to define the drainage network and locate the lakes and reservoirs, the standard
parameterization of PCR-GLOBWB 2 that runs globally uses the drainage network derived from the high
resolution 30 arc-sec HydroSHEDS (Lehner et al., 2008) combined with 30 arc-sec GTOPO30 (Gesch et al., 1999)
and 1 km Hydro1k (Verdin and Greenlee, 1996, USGS EROS Data Center, 2006), lakes taken from GLWD
(Lehner and Döll, 2004) and reservoirs obtained from GranD (Lehner et al., 2011).

**2.3.5 Irrigation and water use module**

In PCR-GLOWB 1 water demand was calculated separately from the hydrology and water availability calculated as a post-processing step by subtracting upstream demand (Wada et al., 2011a,b). In PCR-GLOBWB 2 water use (withdrawal and consumption) is fully integrated. Hereafter, the main features of the irrigation and water use module are described in the following order: water demand, water withdrawal, water consumption and return flows.

*Water demand*

*Irrigation water demand* is calculated based on the crop composition (which changes per month and includes multi-cropping) and the irrigated area per cell. As stated above, these are obtained from MIRCA2000 (Portmann et al., 2010) and the Global Crop Water Model (Siebert and Döll, 2010). In the standard PCR-GLOBWB 2 parameterization the irrigated areas change over time. In want of detailed data, fractions of paddy and non-paddy irrigation, as well as the crop composition per month stay fixed (as obtained from MIRCA2000), while the total irrigated area per cell changes over time and is based on the FAOSTAT (FAO, 2012) reported irrigated areas. Irrigation water demand is computed using the FAO guidelines (Doorenbos and Pruit, 1977, Allen et al., 1998): in case of non-paddy irrigation, water is applied whenever soil moisture falls below a pre-set value and then the soil column is replenished up to field capacity. In case of paddy irrigation, the water level is kept at a water depth of 5 cm above the surface until the late crop development stage ($\sim$ 20 days) before the harvest. After that, no irrigation is applied anymore such that the water level is allowed to drop to zero under infiltration and evaporation (Wada et al., 2014). The net irrigation demand is augmented to account for limited irrigation efficiency and losses. In order to obtain irrigation water demand including losses, i.e. gross irrigation demand, net irrigation water demand is multiplied with $(1 + f_I)$, with $f_I$ a country-specific loss factor obtained from Rohwer et al. (2007).

*Non-irrigation water demand* covers three sectors, industry, households and livestock. For each of these sectors, the gross demand and net demand are prescribed to the model. The calculation of net non-irrigation water demand, which varies with time, follows methods developed by Wada et al (2014). We refer to Wada et al. (2014) for an extensive description. Trends in water demand are prescribed on an annual basis as a function of population, electricity demand and gross domestic product (GDP) per capita. In addition, domestic water demand exhibits a seasonal variation on the basis of temperature. Domestic and industrial gross water demand is calculated from net water demand using a country-specific recycling ratio $RC$ (based on development stage or GDP per capita and additionally access to domestic water demand): gross = net/(1-$RC$). This takes into account that much of the domestic and industrial water is not consumed but returned as surface water. For livestock, the return flow is assumed to be zero, meaning all water is consumed.


*Water withdrawal*

The water withdrawal estimation is based on the work by de Graaf et al. (2014) and Wada et al. (2014). In PCR-
GLOBWB 2 water withdrawal is set equal to gross water demand (summed over all the sectors) unless sufficient
water is not available. In that case, water withdrawal is scaled down to the available water and then allocated
proportionally to gross water demand per sector. Thus, no allocation preference is available in the standard
parameterization of PCR-GLOBWB 2.

Water can be abstracted from three sources: surface water, groundwater (fossil and non-fossil) and desalinated
water. The latter is prescribed (Wada et al., 2011a), while the fractions of the other two sources are determined as a
function of their relative abundance. Groundwater and surface water availability are determined based on two-year
running means of groundwater recharge and river discharge respectively, thus keeping track of the prevalence of
local resources and their temporal change (de Graaf et al., 2014). These fractions determine on a monthly basis
from which source water is abstracted. Surface water withdrawal is ceased if river discharge falls below 10% of
the long-term average yearly discharge under naturalized flow conditions (determined by running the model
without withdrawal). If, for some reason, the surface water amount is insufficient, the model falls back on
groundwater to meet the resulting gap. Groundwater is first abstracted from the renewable groundwater storage,
and if this is not present, non-renewable groundwater is abstracted. The amount of groundwater that can be
abstracted is, however, capped by the groundwater pumping capacity which is based on data by IGRAC GGIS
database. The described dynamic allocation scheme is not always in line with local preferences or the
infrastructure. However, there is a possibility to use fractions of groundwater and surface water withdrawal
reported in the literature. For urban areas, we rely on the data set of McDonald et al. (2014) that states whether a
surface water distribution infrastructure is available. If this is the case, industrial and domestic water withdrawals
are mainly taken from surface water before abstracting groundwater. If surface water infrastructure is limited,
groundwater source is prioritized (see e.g. Erkens and Sutanudjaja, 2015). For urban areas that are not in the
McDonald (2014) data set, we give preference to the dynamic allocation scheme. For irrigation, we use the ratios
supplied by Siebert et al. (2010) in regions where they are said to be reliable. In regions where they are not fully
reliable, we take the average ratio provided by Siebert et al. (2010) and the one provided by the dynamic allocation
scheme. For regions where the data of Siebert (2010) are not reliable (i.e., extrapolated data), we give preference to
the dynamic allocation scheme.

Moreover, we cannot assume that all the water demand is supplied from surface water and groundwater resources
in the same cell. Ideally, data about local water redistribution networks and inter-basin transfers should be used to
define surface water and groundwater service areas. Unfortunately, this information is not available at the global
scale. Therefore, in our current parameterization of PCR-GLOBWB 2, we pool water availability of desalinated
and surface water over zones of approximately 1 arc-degree by 1 arc-degree size that are truncated by country
borders if applicable. For groundwater, 0.5 arc-degree zones are used. The downside of the current scheme is that a
cell does not always have access to its nearest water resource if this lies outside its prescribed service area.

*Water consumption and return flows*

In case of irrigation, all the withdrawn water is applied to the soil (non-paddy) or the water level on the field
(paddy). Part of that water is lost by transpiration and part by soil and open water evaporation. Transpiration and
evaporation together make up the irrigation water consumption. The remaining part of irrigated water is lost by
percolation and contributes to groundwater recharge as return flow. Irrigation efficiency (not including conveyance
losses) could also be calculated after the fact by the difference between withdrawal and transpiration. In case of
domestic and industrial water use, water consumption depends on the recycling ratio RC and equals
withdrawal×(1-RC), while withdrawal×RC constitutes return flow. All return flow is added to the surface water.
For livestock, the consumption is set equal to the withdrawal and no return flow is assumed.

## 2.4 Model code

The original PCR-GLOBWB version 1 (van Beek et al., 2011) was written in the PCRaster scripting language. PCRaster (Wesseling et al., 1996) is a high-level programming language that started as a dynamic raster-based Geographical Information System (GIS) and is tailored to spatiotemporal modelling for environmental and earth science applications. The generic nature of PCRaster with its many pre-existing built-in hydrological functions and its syntax that reads like pseudo-code, generally results in concise model codes, short development times and limited programming errors. Karssenberg et al. (2010) developed a PCRaster Python package such that PCRaster functions, implemented in C++, can also be called via Python (http://www.python.org/). Using PCRaster Python makes it possible for students and beginner modellers to contribute to the model quickly, while it allows experts to be more productive and focus on the science rather than on the programming language syntax. Realising the aforementioned advantages, PCR-GLOBWB, particularly starting from this version 2, has been rewritten in the Python scripting language.

To allow for exchanges of model components and, therefore, evaluate different model configurations, a component-based development approach (e.g Argent, 2004; Castronova and Goodall, 2010) was followed while developing the PCR-GLOBWB 2 model code. Each of the PCR-GLOBWB scientific modules described in section 2.3 is implemented in a separate Python class that needs to implement initialization and update methods. The latter designates changes of states and fluxes per time step. Each of module is initialized and executed by iteratively calling the update method via a main model script.

To run the model a so-called initialization file or configuration file is used (with extension .ini). In this file the following aspects are defined: the spatial and temporal domain, the time step, the settings of the different modules (e.g. which surface water routing, human water use or not etc.) and the locations and names of the parameter files and forcing files. PCR-GLOBWB 2 uses NetCDF files for most input and all output, thus making it easier to exchange data with other scientists and use existing tools to analyse its output.

PCR-GLOBWB 2 generally runs best under Linux. In order to run PCR-GLOBWB the following additional software needs to be installed: PCRaster version 4, Python versions 2.7 with Python packages numPy and netCDF4 and gdal version 1.8 or higher.

**2.5 Differences between PCR-GLOBWB 1 and 2**

PCR-GLOBWB 2 has the following new capabilities compared to PCR-GLOBWB 1 (cf. van Beek et al., 2011, Wada et al, 2011):

- the model was completely rewritten in PCRaster Python and now has a modular structure,
- the inputs and outputs are in the form of NetCDF files and output can be reported for daily monthly and yearly time steps,
- parameterizations are available at 30 arc-minute and 5 arc-minute resolutions,
- water use (demand, withdrawal, consumption and return flow) is fully integrated,
- distinction is made between paddy and non-paddy irrigation and irrigation follows FAO guidelines,
- three different options for surface water routing are available and a surface water temperature module is fully integrated with the routing scheme,
- it is possible to run surface water routines separately with specific discharge from other sources (e.g. other land surface models),
- PCR-GLOBWB 2 can be coupled to a two-layer transient groundwater model (Sutanudjaja et al., 2014, de Graaf et al., 2017) and to the hydrodynamic models Delft3D Flexible Mesh (Kernkamp et al., 2011) or LISFLOOD-FP (Bates et al., 2010, see Hoch et al., 2017b).

## 3. Model demonstration and evaluation

To test and evaluate the performance of PCR-GLOBWB 2, we ran the model at both 30 arc-minute and 5 arc-minute resolution over the period 1958-2015. We compared the results of both simulations with discharge data from the Global Runoff Data Centre (GRDC, 2014), with total basin water storage estimates from GRACE (Gravity Recovery and Climate Experiment, Wiese, 2015) and with water withdrawal data from the FAO AQUASTAT database (FAO, 2016).

### 3.1 Model run setup

**3.1.1 Parameterization**

We used the standard parameterization (parameters, forcing and their sources in Table A2) of PCR-GLOBWB 2 at 30 arc-minute and 5 arc-minute spatial resolutions to simulate global hydrology at daily resolution over 1958-2015. Outputs were reported as monthly averages. The parameterization was mostly unchanged from that given in van Beek and Bierkens (2009), but newer datasets were used if available, such as the GRAND (Lehner et al., 2011) dataset for reservoirs and MIRCA (Portmann et al., 2010) for crop areas. We stress that no calibration was performed. We ran the model with human water use options turned on and used the travel-time characteristic solution routing option.

### 3.1.2 Forcing

The forcing data set is based on time series of monthly precipitation, temperature and reference evaporation from the CRU TS 3.2 data set of Harris et al. (2014) downscaled to daily values with ERA40 (1958-1978, Uppala et al., 2005) and ERA-Interim (1979-2015, Dee et al., 2011). CRU is specified at 30 arc-minute spatial resolution and directly usable. We used ERA40 and ERA-I results that had been resampled by ECMWFs resampling scheme from their original resolutions (~1.2° and ~0.7°) to 30 arc-minutes first. Here, resampling means a form of spatial downscaling whereby the values of the larger ERA40 and ERA-I grid cells are assigned to the cell centers and then spatially interpolated onto 30 arc-minute grids. Precipitation was temporally downscaled by first applying a threshold of 0.1 mm/day to the ERA daily time series to estimate the number of rain days for ERA. The amount of rainfall below this threshold was proportionally allocated to the rain days. Next, the daily rainfall totals were scaled in order to reproduce the CRU monthly precipitation total using multiplicative scaling. Equally, monthly reference potential evaporation, computed with Penman-Monteith from the CRU data set, was scaled using multiplicative scaling and downscaled to daily data proportional to Hamon (1967) evaporation calculated from daily ERA temperatures. We elected not to calculate Penman-Monteith reference evaporation directly from the ERA40 and ERA-I data, in order to avoid the large calculation times needed to process the required meteorological values. For the air temperature, an additive scaling factor was used. To better simulate snow-dynamics for the 5arc-minute model, the temperature values from CRU were further spatially downscaled to 5 arc-minutes using a temperature lapse-rate derived from the higher-resolution CRU CL 2.0 climatology (New et al., 2002). For areas where the number of stations underlying the CRU data set was found to be small, preference was given to using directly the meteorological data from ERA. The method used to create the forcing data set is described more extensively in van Beek (2008).

### 3.1.3 Spin-up

The large groundwater response times for certain regions (e.g. Niger and Amazon) requires substantial spin-up for the groundwater volumes to be in equilibrium with the current climate. To reach this equilibrium, the model was spun-up using the average climatological forcing over the years 1958–2000 back-to-back for 150 years to reach a dynamic steady state. This spin-up was executed under naturalized condition which means no reservoirs and no human water use.

### 3.1.4 Computation time and parallelization

The models were run on Cartesius, the Dutch national supercomputer (https://userinfo.surfsara.nl/systems/cartesius). Without parallelization, the wall clock time for a one-year global simulation run of the 30 arc-minute model was about one hour. This entails that a one-year global simulation run with the 5 arc-minute model, might result in wall clock times of at least 36 hours. Hence, to speed-up computation, the 5 arc-minute model domain was divided into 53 groups of river basins such that it could be run as 53 separate processes. With this simple parallelization technique, the wall clock time for a one-year simulation run of the 5 arc-minute model reduced to about one hour again. Note that these computation times were obtained for simulations with the travel-time characteristic routing option. Calculation times would have been significantly longer if the kinematic wave routing had been used (e.g. about 6 hours for a one-year 5 arc-minutes global run including parallelization).

### 3.2 Data used for comparison

#### 3.2.1 River discharge

We used discharge stations from GRDC (2014) to compare simulated discharge from PCR-GLOBWB 2 with monthly reported discharge. From all the globally available stations in the database, we selected a subset of stations using the following criteria: 1) allowing a not more than 15% difference in catchment area between PCR-GLOBWB 2 and the area reported with the GRDC discharge station, 2) not more than 1 cell distance between the station location and the nearby location of a river in PCR-GLOBWB 2, 3) at least 1 year of discharge data. This yielded 5363 stations for the 5 arc-minute simulation, 3910 stations for the 30 arc-minute simulation and 3597 stations fulfilling the criteria for both resolutions. The minimum, median and maximum catchment sizes for the GRDC stations at the 5 arc-minute resolution are respectively 29, 2730 and $4.68 \cdot 10^6 \text{km}^2$ and 31, 6560 and $4.68 \cdot 10^6 \text{km}^2$ at the 30 arc-minute resolution. As we jointly compared the performance of both simulations, we used the set of 3597 locations throughout. The average time series length of these stations is equal to 36 years.

#### 3.2.2 Total water storage

We compared total water storage (TWS) as simulated by PCR-GLOBWB 2 with the TWS estimated from GRACE (Gravity Recovery and Climate Experiment) gravity anomalies. We used the GRACE JPL Mascon product PL-RL05M (Wiese, 2015, Watkins et al., 2015, Wiese et al., 2016). Scanlon et al. (2016) suggest that recent developments in mascon (mass concentration) solutions for GRACE have significantly increased the spatial localization and amplitude of recovered terrestrial TWS signals. They also claim that one of the advantages of using the mascon solutions relative to traditional SH (spherical harmonic) solutions is that it makes it much easier for non-geodesists to apply GRACE data to hydrologic problems. Note that although the data of PL-RL05M are represented on a 30 arc-minutes lat-lon grid, they represent the 3x3 arc-degree equal-area zones, which is the actual resolution of JPL-RL05M. We compared trends on

a pixel-by-pixel basis. Given the coarse resolution of GRACE products of about 300 km by 300 km we
compared correlations only for major river basins with an area of 900,000 km$^2$ and up.


**3.2.3 Water withdrawal**

The water withdrawal for a large number of countries is taken from FAO's AQUASTAT database (FAO,
2016). This data is on average reported in every 5 years. We compared simulated water withdrawal per
sector and per water source (surface water and groundwater) with reported values per country and per
reporting period, whenever available.


**3.3 The global water balance simulated at 30 and 5 arc-minutes**

We calculated the main global water balance components from the 30 arc-minute and 5 arc-minute
simulations over the period 2000-2015. The results in Table 1 show that there are some differences
between the two model runs, but values are in the same order of magnitude. The small difference in
precipitation is due to the fact that the area of the land cells is slightly different at the two resolutions.
Differences in evaporation and runoff show that the runoff and evaporation parameterization of PCR-
GLOBWB 2 is not entirely scale-consistent. Differences in evaporation may also be causing the differences
in irrigation water demand which in turn may explain the differences in water withdrawal. Recently,
Samaniego et al. (2017) applied their multiscale parameter regionalization (creating spatially variable
parameter fields) technique (MPR) to PCR-GLOBWB 2 for the Rhine basin, showing that
parameterizations that yield the same hydrological fluxes at different resolutions are possible. However, a
global application of this method to all PCR-GLOWB 2 parameters is not possible yet. Nonetheless, when
comparing the results of both model runs with data reported in the literature, it shows that the global water
balance components are similar to recent assessments (e.g. by Rodell et al., 2015) and groundwater
withdrawal and total withdrawal estimates match those of previous studies (see Table 2).

From Table 1, it can also be seen that there is a negative change in total terrestrial water storage in both
model runs. Table 1 shows that this can only be partly explained by groundwater depletion, which is
localized to certain regions (see also Sect. 3.4.2). Further analysis shows that this change can also be
attributed to the trends in precipitation forcing used, particularly over the tropics.
*Table 1. Global Water balance components and human water withdrawal (in km³/year and mm/year) over*
*the period 2000-2015 as obtained from the 30 arc-minutes and the 5 arc-minute simulations. The numbers*
*are shown to high significance to show the water balance closure. This does not mean that we pretend to*
*know e.g. global discharge with a km³ accuracy (actual accuracy of the large fluxes is more in the order of*
*$10^3$ km³)*

| | | 30 arc-min | | 5 arc-min | |
|---|---|---|---|---|---|
| | | km³/year | mm/year | km³/year | mm/year |
| Global water balance | Precipitation | 107452 | 808 | 107495 | 811 |
| | Desalinated water use | 3 | 0.02 | 2 | 0.01 |
| | Runoff | 42393 | 319 | 43978 | 332 |
| | Evaporation* | 65754 | 494 | 63974 | 483 |
| | Change in total water storage | -693 | -5 | -455 | -3 |
| Groundwater budget | Groundwater recharge | 27756 | 209 | 25521 | 193 |
| | Groundwater withdrawal | 737 | 6 | 632 | 5 |
| | Non-renewable groundwater withdrawal (groundwater depletion) | 173 | 1 | 171 | 1 |
| | Renewable groundwater withdrawal | 564 | 4 | 460 | 3 |
| Withdrawal by sector | Agricultural water withdrawal (irrigation + livestock) | 2735 | 21 | 2309 | 17 |
| | Domestic water withdrawal | 380 | 3 | 314 | 2 |
| | Industrial water withdrawal | 798 | 6 | 707 | 5 |
| Withdrawal by source | Total water withdrawal | 3912 | 29 | 3330 | 25 |
| | Surface water withdrawal | 3172 | 24 | 2697 | 20 |
| | Desalinated water use | 3 | 0.02 | 2 | 0.01 |
| | Groundwater withdrawal | 737 | 6 | 632 | 5 |

* Includes consumptive water use for livestock, domestic and industrial sectors


*Table 2. Groundwater withdrawal and total water withdrawal as compared to other studies (in km³/year)*

| | Source | Year | Value (km³/year) |
|---|---|---|---|
| Groundwater withdrawal | Wada et al. (2010) (from the IGRAC database) | 2000 | 734 (±87) |
| | Döll et al. (2012) | 1998-2002 | 571 |
| | Döll et al. (2014) (their Table 2). | 2003-2009 | 690-888 |
| | Döll et al. (2014) (their Table 6). | 2000-2009 | 665 |
| | Pokhrel et al. (2015) | 1998-2002 | 570 (±61) |
| | Hanasaki et al. (2018) | 2000 | 789 (±30) |
| | This study (5 arc-minutes) | 2000-2015 | 632 |
| Total water withdrawal | Vörösmarty et al. (2005) | 1995-2000 | 3560 |
| | Oki and Kanae (2006) | contemporary | 3800 |
| | Döll et al. (2012) | 1998-2002 | 4340 |
| | Döll et al. (2014) (their Table 2) | 2003-2009 | 3000-3700 |
| | FAO (2016) | 2010 | 3583 |
| | Hanasaki et al. (2018) | 2000 | 3628 (±75) |
| | This study (5 arc-minutes) | 2000-2015 | 3330 |



### 3.4 Evaluation of the 30 and 5 arc-minute simulations

### 3.4.1 Discharge

When evaluating the simulated discharge with discharge observations from GRDC, we used the monthly values and calculated three different measures. The first one is the correlation coefficient between monthly simulated and observed GRDC time series, which is a measure of reproducing correct timing of high and low discharge. A correlation coefficient of 1 indicates perfect timing. The second measure is the Kling-Gupta efficiency coefficient or KGE (Gupta et al., 2009) which equally measures bias, differences in amplitude and differences in timing between monthly simulated and observed GRDC time series. The KGE varies between 1 and minus infinity, where 1 means a perfect fit in terms of bias, amplitude and timing. The last metric is the anomaly correlation, i.e. the correlation between monthly time series after the seasonal signal (climatology) has been removed. This statistic measures the ability of the model to correctly simulate timing of seasonal and the inter-annual anomalies from the yearly climatology. This is to test if the model is able to capture the monthly scale and inter-annual anomalies in discharge (i.e. on the monthly scale) when the dominant seasonal trend is removed from observations and simulations. An anomaly correlation of 1 indicates perfect characterization of inter-annual anomalies and values below 0 indicate a lack thereof.

Figure 2 shows maps of the correlation coefficients for the GRDC stations considered and Figure 3 shows histograms of correlation and KGE values. Both figures show that the evaluation results of the 5 arc-minute simulation are generally better than those of the 30 arc-minute simulation. For the 30 arc-minute model, the number of catchments with KGE > 0, 0.3 and 0.6 are equal to 48%, 26% and 7% of the total catchments respectively. For the 5 arc-minute model, these values are respectively equal to 63%, 40% and 12% of the total catchments. Note that for both runs the standard parameterization was used. Possible explanations for the better performance of the 5 arc-minute run are: a better delineation of the shape of the basins, particularly the smaller ones, a better characterization of basin relief and the drainage network, more accurate sub-grid parameterization of soil and land cover due to a smaller scale-gap that needs to be overcome, better estimates of the basin storage and better snow dynamics due to the downscaling of temperature to 5 arc-minute resolution. The KGE values are less favourable than the correlation coefficients. This is mostly due to biases in runoff caused by incorrect meteorological forcing. It is difficult to exactly assess which of these factors are most important in determining the improvement. Inspecting the histograms of correlation and KGE (Figure 3) shows that the improvement is mostly apparent for the smaller sized catchments, which supports the notion that a better delineation of the catchments' shape, topography and drainage network could be the cause. However, disentangling these individual effects would require further study. To investigate the possible effects of better snow dynamics we classified the GRDC stations into stations below 1000 m altitude (above mean sea-level) and those above 1000 m. The GRDC stations above 1000 m are expected to experience precipitation falling as snow during periods of the year. The results in Figure 4 clearly show that the improvement is larger for the higher GRDC stations, This supports the explanation that better snow dynamics due to temperature lapsing in combination with a better resolved digital elevation model is partly responsible for the superior results at 5 arc-minutes. We

also investigated if improvements were notably different between climate zones, by separately calculating
KGEs for GRDC stations in the Köppen-Geiger zones A (Tropical), B (Desert), C (Temperate) and D
(Continental). The results (not shown) show that the improvement is equally visible for climate zones A, B
and C and less so for D (continental). Without further analysis this is difficult to explain. Note however that
the continental climate zone is somewhat under-represented in the GRDC dataset due to the low
measurement densities over Russia, although it is well represented in the U.S. So, it may be that the global
improvements shown in Figure 3 are somewhat positively biased.

The maps of correlations (Figure 2) show the best results in Europe and North America where the
meteorological forcing is generally more accurate as a result of more data used in the re-analysis products
and higher station availability in the CRU data set. Also, monsoon-dominated basins are well simulated due
to the strong seasonal nature of both forcing and related discharge. The improvement of the 5 arc-minute
simulation over the 30 arc-minute simulation in Europe is mostly seen in the Alps and the Norwegian
mountains. This reflects the fact that topography and thus snow dynamics is better represented at higher
resolution as shown in Figure 4. The least accurate results are obtained for some of the African rivers, in
particular the Niger where the groundwater recession coefficients are probably over-estimated and inland
delta evaporation is under-estimated, for some rivers in the Rocky Mountains, which may be the result of
errors in snow dynamics and for continental Eastern Europe, which is most likely explained by an over-
estimation of the groundwater recession constants.


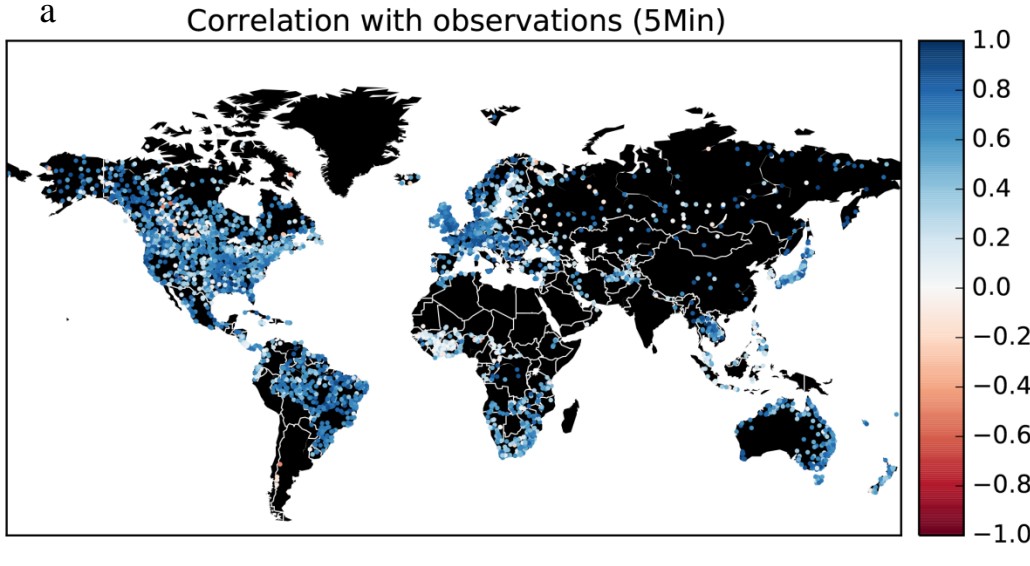

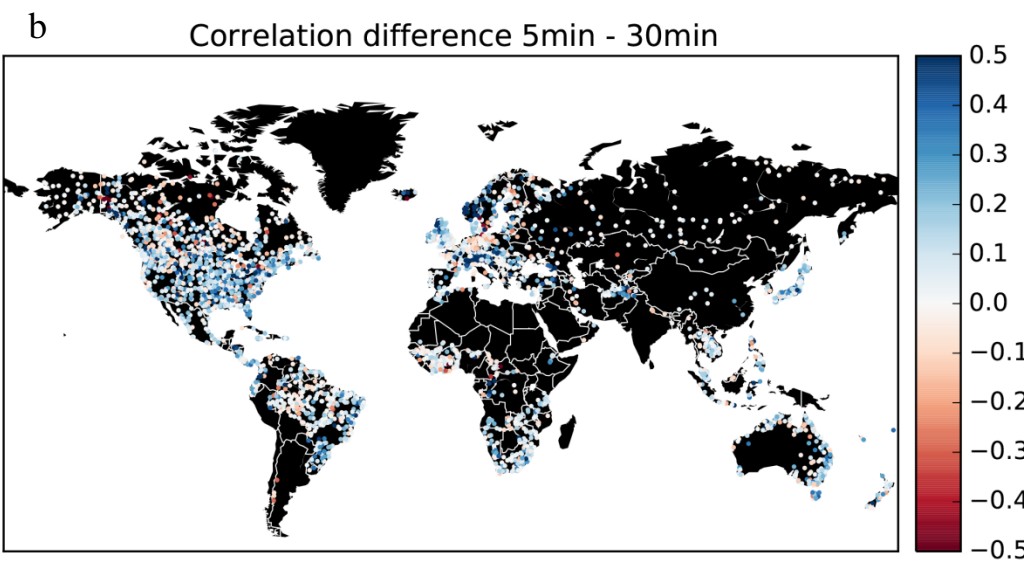

Figure 2. *Maps of correlation between simulated and observed discharge time series for 3597 GRDC discharge stations; a. results for the 5 arc-minutes simulation; b. difference between results for 5 arc-minutes and 30 arc-minutes simulation.*

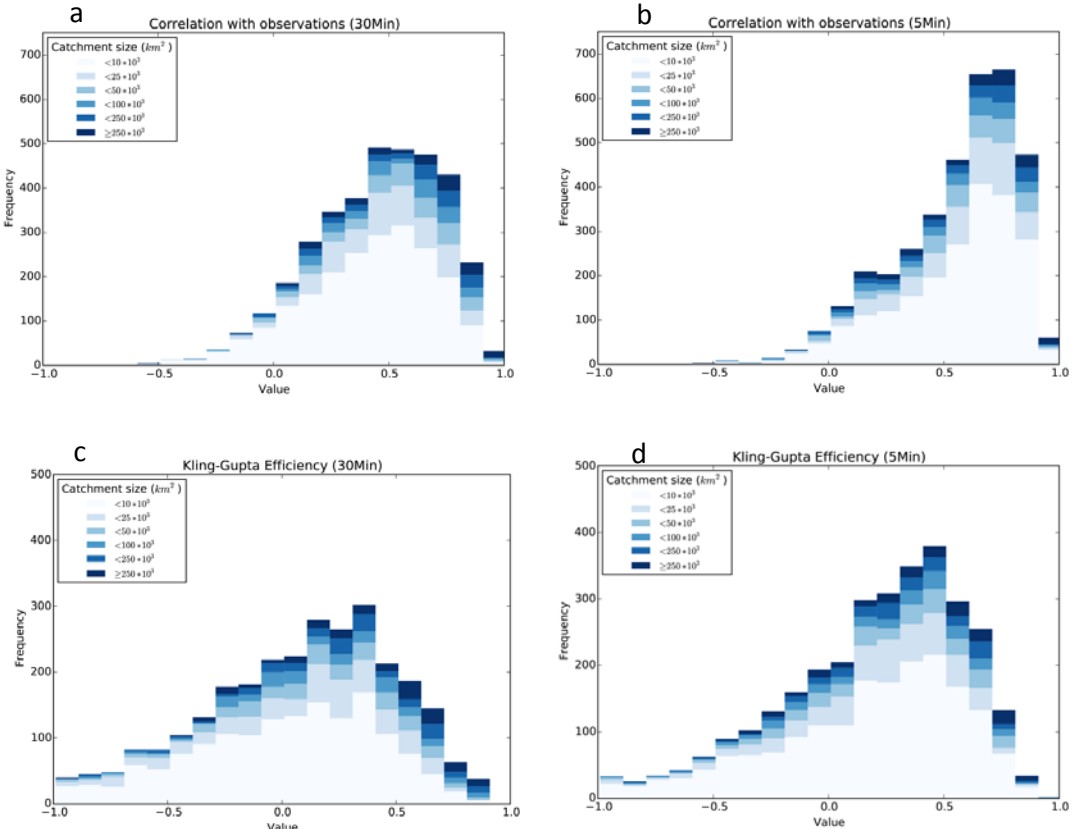

*Figure 3. Histograms of evaluation statistics showing the correlation and Kling-Gupta efficiency (KGE) values for the simulated discharge for the 30 arc-minutes and the 5 arc-minute simulations based on 3597 GRDC discharge stations, a. correlation 30 arc-minute simulation, b. correlation 5 arc-minute simulation, c. KGE 30 arc-minute simulation, d. KGE 5 arc-minute simulation, note: the percentage catchments with KGE < -1 are 21% and 12% for 30 and 5 arc-minutes respectively.*

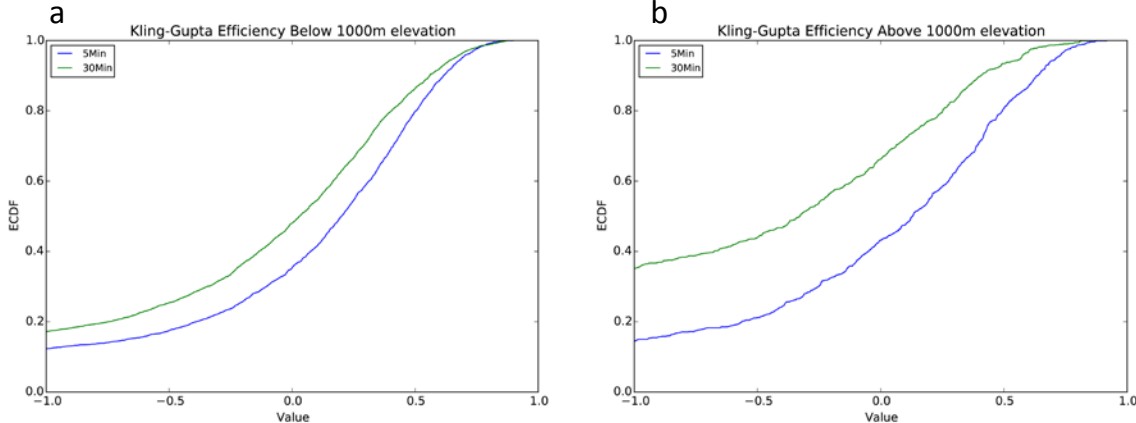

699

*Figure 4. Cumulative frequency distributions of Kling-Gupta efficiency (KGE) values for GRDC stations that are*

*positioned below (a) and above (b) 1000 m a.m.s.l. It can be expected that for the stations above 1000 m, the*

*upstream area is influenced by snow dynamics.*

The histograms of the anomaly correlation are shown in Figure 5. The anomaly correlations are generally lower

than the correlations, showing that seasonality explains part of the skill in many regions where seasonal variation is

dominant when compared to intra-annual or inter-annual variability. Clearly, the 5 arc-minute results are much

better than those of the half-degree simulation, indicating a higher skill with regard to capturing inter-annual

anomalies. Figure 6 shows a map of the difference between the anomaly correlation and the correlation for the 5

arc-minute case. This map shows that there are some regions where the anomaly correlation is better than the

correlation (blue colours), e.g. snow-dominated regions in Canada and the Niger basin. These are catchments

where the model has difficulty reproducing the correct seasonality as a result of errors in snow dynamics (Canada)

or groundwater dynamics (Niger). Also, in case of the Niger River, not representing the inner delta flooding and

resulting high evaporation may be the cause of poor seasonal timing of discharge.

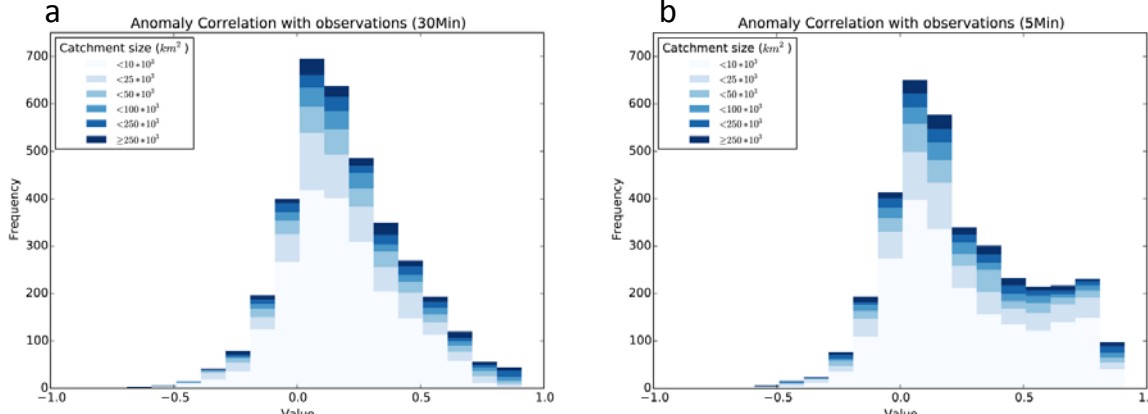


*Figure 5. Histograms of evaluation statistics showing the anomaly correlation for the simulated discharge for the*

*30 arc-minutes and the 5 arc-minute simulations based on 3597 GRDC discharge stations, a. anomaly correlation*

*half arc-degree simulation, b. anomaly correlation 5 arc-minute simulation.*



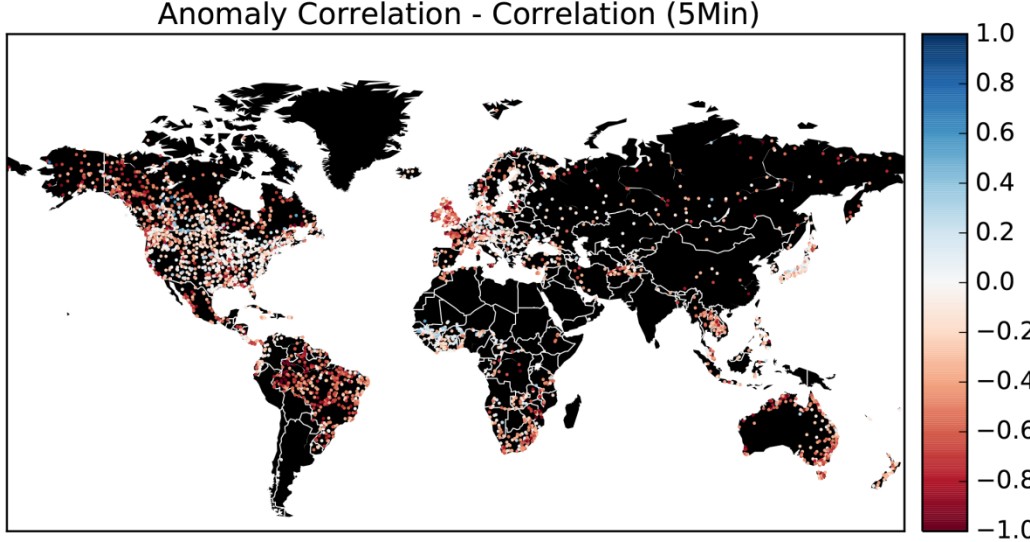


*Figure 6. Map showing for the 5 arc-minute run the difference between the correlation and the anomaly*

*correlation between simulated and observed discharge time series for 3597 GRDC discharge stations, negative*

*values mean that the correlation is higher than the anomaly correlation.*


### 3.4.2 Total water storage

Figure 7 compares the trends in 5 arc-minute simulated total water storage (TWS) with those from GRACE, estimated as the average change in m/year over the period 2003-2015. Generally, the PCR-GLOBWB 2 simulation is able to capture major groundwater depleted regions as suggested by GRACE, such as those in the Central Valley aquifer, the High Plains aquifer, the North China Plain aquifer, as well as parts of the Middle East, Pakistan and India. For these regions, the absolute rates of TWS change (i.e. TWS declines) of PCR-GLOBWB 2 are generally larger, while the spatial pattern in the GRACE map tends to be smoother. This is mainly due to the lower resolution and spatial averaging used in the GRACE product, as well as the fact that the current PCR-GLOBWB 2 simulation does not include lateral groundwater flow between cells. In the polar regions where GRACE estimates mass loss due to melting glaciers and ice sheets, PCR-GLOBWB 2 simulates accumulation as a result of lack of a glacier parameterization. Finally, there are some clear differences over the Amazon and some parts of Africa. A possible explanation are errors in meteorological forcing data, which is not very accurate in these parts, but also problems with the over-estimation of PCR-GLOBWB's groundwater response times in these regions which therefore fail to be sufficiently sensitive to recent changes in terrestrial precipitation.

Further analyses were conducted at the basin-scale resolution, where both TWS time series of PCR-GLOBWB 2 and GRACE *JPL-RL05M* were averaged over a river basins areas map derived from the 5 arc-minute PCR-GLOBWB drainage network. We identified all river basins with sizes larger than 900,000 km$^2$, which is similar to the GRACE resolution. Smaller river basins were merged to the nearest river basins or grouped together. For the remaining map of large basins, the correlations between PCR-GLOBWB 2 and GRACE basin-average monthly and annual TWS time series were calculated. Monthly correlation provides information about PCR-GLOBWB's ability to correctly time TWS seasonal variability (with a value equal to 1 for perfect timing), while the correlation for annual time series measures inter-annual variability.

The results in Figure 8 show that PCR-GLOBWB 2 is able to capture GRACE's TWS seasonality for most basins around the world, with the exception of some cold regions in high latitudes (e.g. the Yukon River basin, Iceland). This shortcoming is most likely due to the lack of a proper representation of glacier and ice processes in PCR-GLOBWB 2. As expected, the correlation values for inter-annual time series are generally lower than the ones for monthly time series. There are some areas with negative correlation values, such as the Amazon, Niger and Nile river basins. Apart from the uncertainty in the GRACE signal, these deficiencies may be related to errors in model forcing and structural errors such as errors in the groundwater response time and the effects of wetlands that have not been represented sufficiently well.

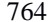

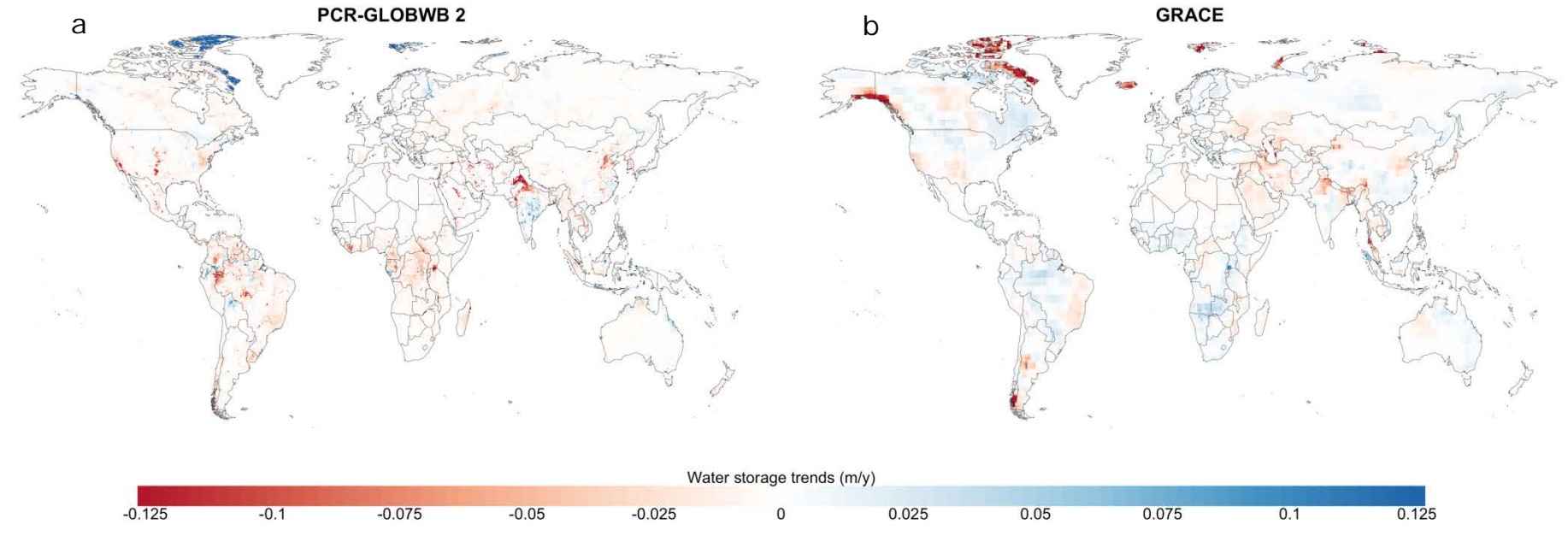

*Figure 7. Comparison of PCR-GLOBWB 2 total water storage trends (m/year) with those estimated with GRACE over the period 2003-2015. a. TWS trends simulated with PCR-*
*GLOBWB at 5 arc-minutes resolution (~10 km at the equator). Negative values indicate declining TWS (e.g. groundwater depleted regions). b. TWS trends obtained based on the*
*GRACE JPL PL-RL05M Mascon product. The GRACE data were resampled to the resolution of 30 arc-minutes, but they actually represent the 3 x 3 arc-degree (~300 km x 300*
*km) area, which is the native resolution of the GRACE signal.*

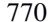

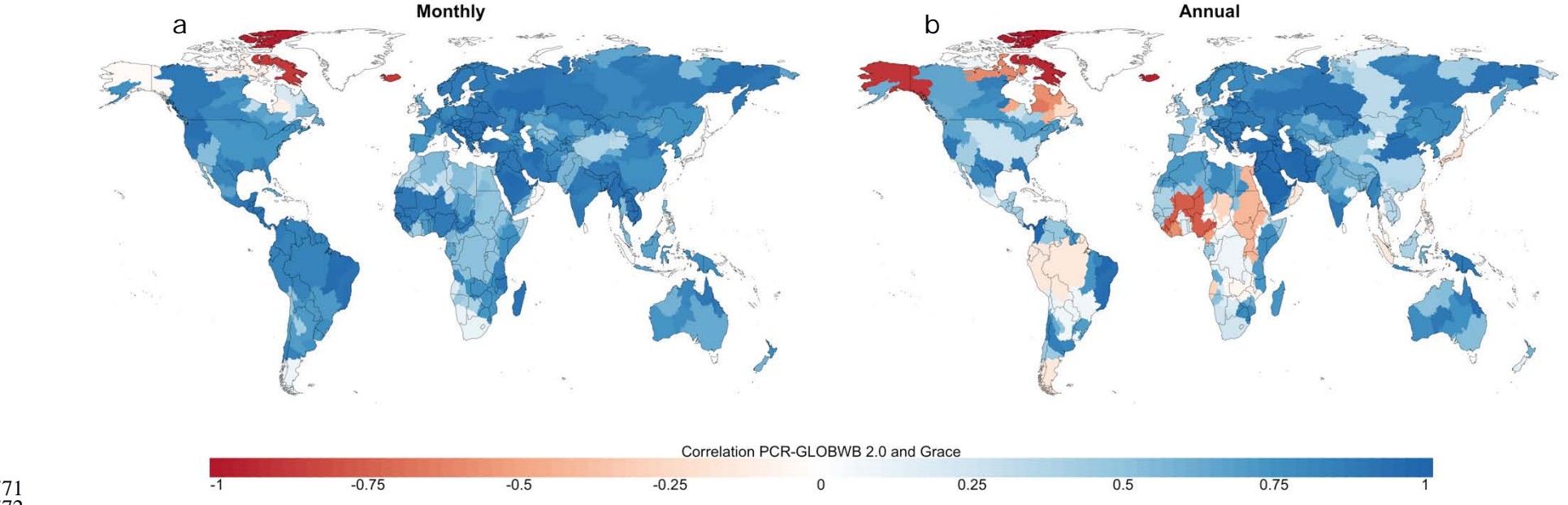

*Figure 8. a. Correlation between monthly TWS time series simulated PCR-GLOBWB 2 and the GRACE JPL PL-RL05M Mascon product over the period 2003-2015. b.*

*Comparison of annual TWS series (inter-annual variability). Comparison is only done for the larger basins over 900,000 km2, conform the 3x3 arc-degree resolution of GRACE.*

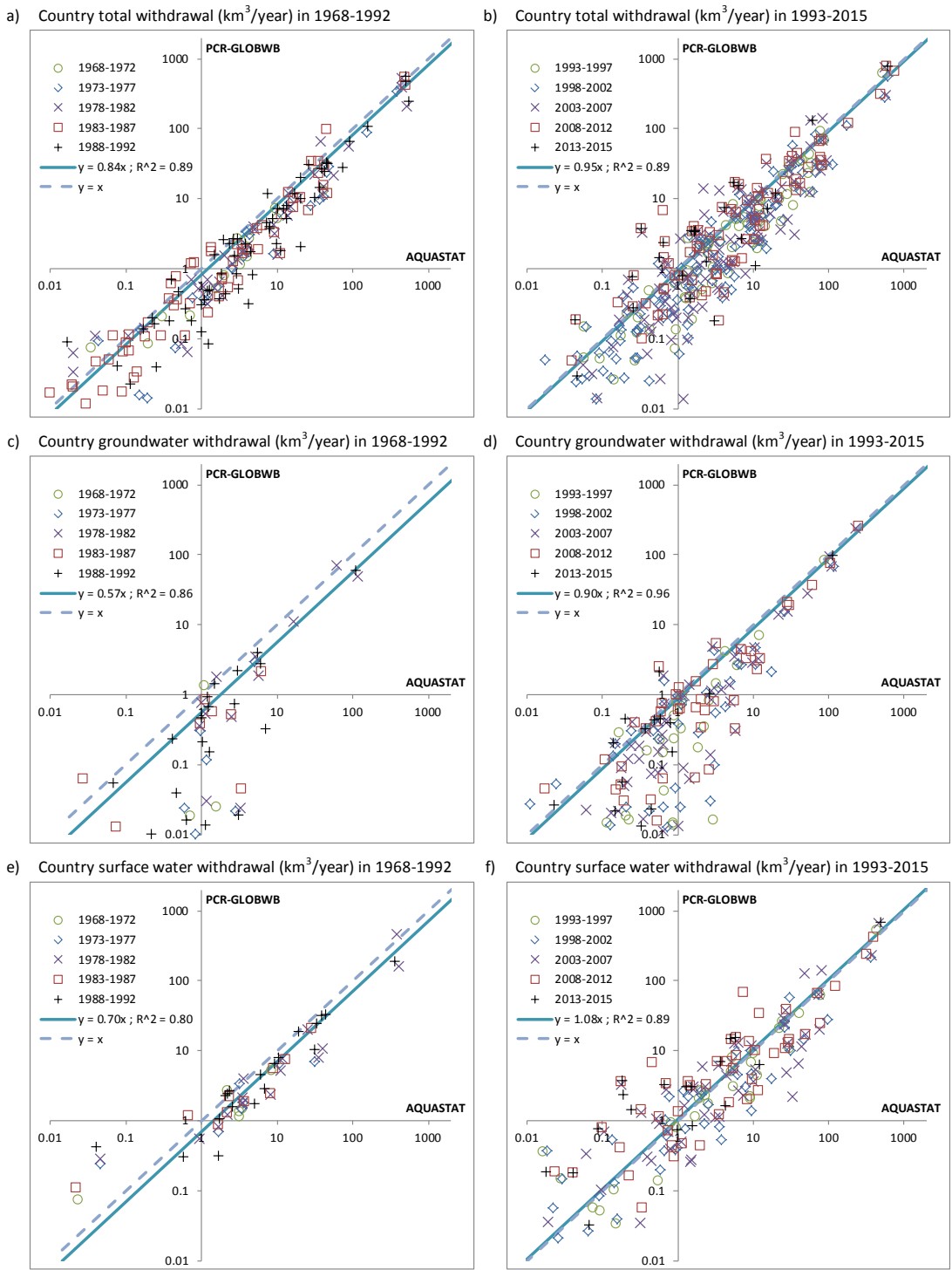


Fig. 9: Country water withdrawal (km³/year) by source, evaluation of simulations with PCR-GLOBWB 2 with reported values in AQUASTAT (FAO, 2016). The scatterplots on the left (a, c, e) are for the period 1968-1992, while the right ones (b, d, f) are 1993-2015. The uppermost plots (a, b) are for total water withdrawal, the middle ones (c, d) are groundwater withdrawal, and the lowermost charts (e, f) are surface water withdrawal. Regression coefficient based on regression to non-log transformed data with intercept kept zero.

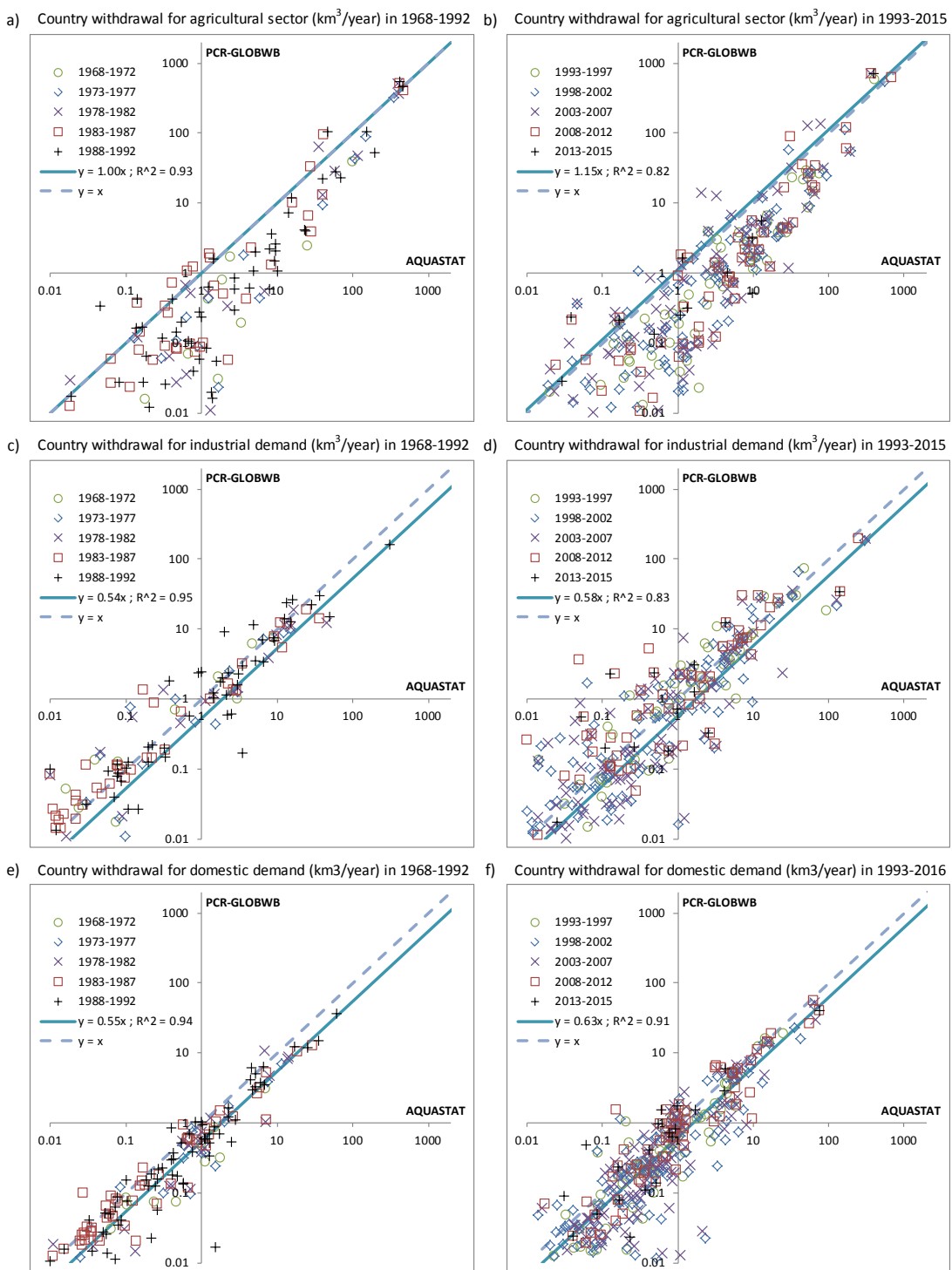

781

*Fig. 10: Country water withdrawal (km3/year) by sector, evaluation of simulations with PCR-GLOBWB 2*
*with reported values in AQUASTAT (FAO, 2016). The scatterplots on the left (a, c, e) are for the period 1968-*
*1992, while the right ones (b, d, f) are 1993-2015. The uppermost plots (a, b) are for withdrawal for*
*agricultural purpose, the middle ones (c, d) are industrial withdrawal, and the lowermost charts (e, f) are*
*domestic. Regression coefficient based on regression to non-log transformed data with intercept kept zero.*


**3.4.3 Water withdrawal**

We compared simulated water withdrawal data from PCR-GLOBWB 2 with reported withdrawal data per
country from AQUASTAT (FAO, 2016). The results are shown subdivided per source (Figure 9) and per
sector (Figure 10).. Total water withdrawal and surface water withdrawal are simulated reasonably well ($R^2$
between 0.84 and 0.96 and regression slopes between 0.70 and 1.08). However, groundwater withdrawal is
underestimated for the smaller water users. A likely explanation for this is occasional groundwater
withdrawal by farmers during dry periods in areas that have not been mapped as irrigated crops in MIRCA,
such as grasslands in e.g. Germany and the Netherlands, while this groundwater withdrawal is reported in
AQUASTAT.

When looking at water withdrawal per sector, results are mixed. The largest agricultural water users are well
captured, but the smaller ones are clearly underestimated. This is related to the fact that in many regions of the
smaller water use countries, water is used for irrigation only occasionally during dry summers, while these
areas are not mapped as irrigated crops in MIRCA. Also, many of these countries use irrigation technology
that is not part of MIRCA, e.g. subsurface drainage by artificially high surface water levels such as in a
number developed delta regions in the world. However, even though these smaller countries are not well
represented, PCR-GLOBWB 2 is still able to capture the big water users, which have a significant impact on
the water cycle and are most important for global scale analyses.

Both industrial and domestic water withdrawals are underestimated. The underestimation of industrial water
withdrawal is partly caused by the fact that we do not include water withdrawal for thermo-electric cooling of
power plants. The underestimation of domestic water withdrawal comes from the fact that we assume that the
priority of water allocation is proportional to demand. This means that in times of shortage, water withdrawal
is reduced with an equal percentage for agriculture, industry and domestic use. In many countries however,
there is a priority series, whereby domestic demand is first met, industrial demand next and agricultural
demand comes last. As a result, we underestimate domestic water withdrawal and it also partly causes the
underestimation of industrial water withdrawal. This is corroborated by plotting gross water demand (which
would be withdrawal if no shortage would occur) against AQUASTAT data. These plots (not shown here)
result in a regression slopes of 0.68-0.75 for industrial demand and 0.78-0.92 for domestic demand. These
results thus reveal that the water allocation scheme of PCR-GLOBWB 2 should be further improved.


## 4. Conclusions and future work

We presented the most recent version of the open source global hydrology and water resources model PCR-GLOBWB. This version, PCR-GLOBWB 2, has a global coverage at 5 arc-minute resolution. Apart from the higher resolution, the new model has an integrated water use scheme, i.e. every day sector specific water demand is calculated, resulting in groundwater and surface water withdrawal, water consumption and return flows. Dams and reservoirs from the GranD database (Lehner et al., 2011) are added progressively according to their year of construction. PCR-GLOBWB 2 has been rewritten in Python and uses PCRaster-Python functions (Karssenberg et al., 2007). It has a modular structure, which makes the replacement and maintenance of model parts easier. PCR-GLOBWB 2 can be dynamically coupled to a global 2-layer groundwater model (de Graaf et al., 2017; Sutanudjaja et al., 2014; Sutanudjaja et al., 2011) and a one-way coupling to hydrodynamic models for large-scale inundation modelling (Hoch et al., 2017b) is also available.

Comparing the 5 arc-minute with 30 arc-minute simulations using discharge data we clearly find an improvement in the model performance of the higher resolution model. We find a general increase in correlation, anomaly correlation and KGE, indicating that the higher resolution model is better able to capture the seasonality, inter-annual anomalies and the general discharge characteristics. Also, PCR-GLOBWB 2 is able to reproduce trends and seasonality in total water storage as observed by GRACE for most river basins. It simulates the hotspots of groundwater decline that around in GRACE as well. Simulated total water withdrawal matches reasonably well with reported water withdrawal from AQUASTAT, while water withdrawal by source and sector provide  mixed results.

Future work will concentrate on further improving the water withdrawal and water allocation scheme, developing a full dynamic (two-way) coupling with hydrodynamic models, developing 5 km and 1 km resolution (or higher) parameterizations of PCR-GLOBWB 2 using scale-consistent parameterizations (e.g. using MPR, Samaniego et al., 2017), incorporating a crop growth model and solving the full surface energy balance. Other foreseeable developments are using the model in probabilistic settings and in data-assimilation frameworks.


## 5. Code and data availability


PCR-GLOBWB 2 is open source and distributed under the terms of the GNU General Public License version
3, or any later version, as published by the Free Software Foundation. The model code is provided through a
Github repository: https://github.com/UU-Hydro/PCR-GLOBWB_model (Sutanudjaja et al., 2017a,
https://doi.org/10.5281/zenodo.595656). This keeps users and developers immediately aware of any new
revisions. Also, it allows developers to easily collaborate, as they can download a new version, make changes,
and suggest and upload the newest revisions. The configuration ini-files for the global 30 arc-minutes and
5arc-minute models and the associated model parameters and input files are provided on
https://doi.org/10.5281/zenodo.1045338 (Sutanudjaja et al., 2017b). Development and maintenance of the
official version (main branch) of PCR-GLOBWB 2 is conducted at the Department of Physical Geography,
Utrecht University. Yet, contributions from external parties are welcome and encouraged. For news on latest
developments and papers published based on PCR-GLOBWB 2 we refer to http://www.globalhydrology.nl
and for the underlying PCRaster-Python code to http://pcraster.geo.uu.nl.


## Acknowledgements

We thank Utrecht University and various grants and projects that directly or indirectly contributed to the
development of PCR-GLOBWB 2. The authors are very grateful to all the contributors (as acknowledged in
the references) who provided the data sets used in this study. We acknowledge the Netherlands Organisation
for Scientific Research (NWO) for the grant that enabled us to use the national super computer Cartesius with
the help of SURFsara Amsterdam. The authors thank two anonymous reviewers for their constructive
comments and suggestions that helped to improve the manuscript. We are also grateful to the Editors for their
efficient handling of the review process.

# Appendix

**Table A1 - List (non-exhaustive) of state and flux variables defined in PCR-GLOBWB**

| Description | Symbol | Unit |
|---|---|---|
| Interception storage | $S_{int}$ | m |
| Snow cover/storage in water equivalent thickness (excluding liquid part $S_{slq}$) | $S_{swe}$ | m |
| Liquid/melt water storage in the snow pack | $S_{slq}$ | m |
| Upper and lower soil storages | $S_1$ and $S_2$ | m |
| Surface water storage (lakes, reservoirs, rivers and inundated water) | $S_{wat}$ | m |
| groundwater storage (renewable part) | $S_3$ | m |
| fossil groundwater storage (non-renewable) | $S_{nrw}$ | m |
| total groundwater storage = $S_3 + S_{nrw}$ | $S_{gwt}$ | m |
| total water storage thickness = $S_{int} + S_{swe} + S_{slq} + S_1 + S_2 + S_{gwt}$ | TWS | m |
| potential evaporation | $E_{pot}$ | m.day$^{-1}$ |
| evaporation flux from the intercepted precipitation | $E_{int}$ | m.day$^{-1}$ |
| evaporation from melt water stored in the snow pack | $E_{slq}$ | m.day$^{-1}$ |
| bare soil evaporation | $E_{soil}$ | m.day$^{-1}$ |
| transpiration from the upper and lower soil stores | $T_1$ and $T_2$ | m.day$^{-1}$ |
| total land evaporation = $E_{pot} + E_{int} + E_{slq} + E_{soil} + T_1 + T_2$ | $E_{land}$ | m.day$^{-1}$ |
| surface water evaporation | $E_{wat}$ | m.day$^{-1}$ |
| total evaporation = $E_{land} + E_{wat}$ | $E_{tot}$ | m.day$^{-1}$ |
| direct runoff | $Q_{dr}$ | m.day$^{-1}$ |
| interflow, shallow sub-surface flow | $Q_{sf}$ | m.day$^{-1}$ |
| baseflow, groundwater discharge | $Q_{bf}$ | m.day$^{-1}$ |
| specific runoff from land | $Q_{loc}$ | m.day$^{-1}$ |
| local change in surface water storage | $Q_{wat}$ | m.day$^{-1}$ |
| total specific runoff | $Q_{tot}$ | m.day$^{-1}$ |
| routed channel (surface water) discharge | $Q_{chn}$ | m$^3$.sec$^{-1}$ |
| net fluxes from the upper to lower soil stores | $Q_{12}$ | m.day$^{-1}$ |
| net groundwater recharge, fluxes from the lower soil to groundwater stores | RCH = $Q_{23}$ | m.day$^{-1}$ |
| surface water infiltration to groundwater | Inf | m.day$^{-1}$ |
| desalinated water withdrawal | $W_{sal}$ | m.day$^{-1}$ |
| surface water withdrawal | $W_{wat}$ | m.day$^{-1}$ |
| renewable groundwater withdrawal | $W_3$ | m.day$^{-1}$ |
| non-renewable groundater withdrawal (groundwater depletion) | $W_{nrw}$ | m.day$^{-1}$ |
| total groundwater withdrawal = $W_3 + W_{nrw}$ | $W_{gwt}$ | m.day$^{-1}$ |
| water withdrawal allocated for irrigation purpose | $A_{irr}$ | m.day$^{-1}$ |
| water withdrawal allocated for livestock demand/sector | $A_{liv}$ | m.day$^{-1}$ |
| water withdrawal allocated for agricultural sector = $A_{irr} + A_{liv}$ | $A_{agr}$ | m.day$^{-1}$ |
| domestic water withdrawal | $A_{dom}$ | m.day$^{-1}$ |
| industrial water withdrawal | $A_{ind}$ | m.day$^{-1}$ |

**Table A2 - List of model inputs and parameters**

| Description | Symbol | Unit | References/sources |
|---|---|---|---|
| Upper and lower soil store parameters: | | | FAO (2007) soil map; van Beek and Bierkens (2009) |
| - Soil thickness | $Z_1$ and $Z_2$ | m | |
| - Residual soil moisture content | $\theta_{r-1}$ and $\theta_{r-2}$ | $m^3.m^{-3}$ | |
| - Soil moisture at saturation | $\theta_{s-1}$ and $\theta_{s-2}$ | $m^3.m^{-3}$ | |
| - Soil water storage capacity per soil layer: $SC = Z / (\theta_s - \theta_r)$ | $SC_1$ and $SC_2$ | m | |
| - Soil matric suctions at saturation | $\psi_{s-1}$ and $\psi_{s-2}$ | m | |
| - Exponent in the soil water retention curve | $\beta_1$ and $\beta_2$ | dimensionless | |
| - Saturated hydraulic conductivities of upper and lower soil stores | $K_1$ and $K_2$ | $m.day^{-1}$ | |
| - Total soil water storage capacities = $SC_{upp} + SC_{low}$ | $W_{max}$ | m | |
| Land cover fraction: Land cover areas (including extent of irrigated areas) over cell areas | $f_{lcov}$ | $m^2.m^{-2}$ | GLCC v2.0 map (USGS, 1997); Olson (1994a, 1994b); MIRCA2000 dataset (Portmann et al., 2010), FAOSTAT (2012) |
| Topographical parameters | DEM | m | HydroSHEDS (Lehner et al., 2008); Hydro1k (Verdin and Greenlee, 1996); GTOPO30 (Gesch et al., 1999) |
| - Cell-average DEM | $DEM_{avg}$ | m | |
| - Flood plain elevation | $DEM_{fpl}$ | m | |
| Root fractions per soil layer | $Rf_{upp}$ & $Rf_{low}$ | dimensionless | Canadell et al. (1996); van Beek and Bierkens (2009) |
| Arno scheme (Todini, 1999; Hagemann and Gates, 2003) exponents defining soil water capacity distribution | $\beta_{arno}$ | dimensionless | Canadell et al. (1996), Hagemann et al. (1999); Hagemann (2002); van Beek (2008); van Beek and Bierkens (2009) |
| Ratios of cell-minimum and cell-maximum soil storage to $W_{max}$ | $f_{wmin}$ and $f_{wmax}$ | m | van Beek (2008); van Beek and Bierkens (2009) |


**Table A2 - List of model inputs and parameters (continued)**

| Description | Symbol | Unit | References/sources |
|---|---|---|---|
| Parameters related to phenology | | | Hagemann et al. (1999); Hagemann (2002); van Beek (2008); van Beek and Bierkens (2009) |
| - Crop coefficient | $K_c$ | dimensionless | |
| - Interception capacity | $S_{int\text{-}max}$ | m | |
| - Vegetation cover fraction | $C_v$ | $m^2.m^{-2}$ | |
| Groundwater parameters | | | GLHYMPS map (Gleeson et al., 2014); van Beek (2008); van Beek and Bierkens (2009) |
| - Aquifer transmissivity | $KD$ | $m^2.day^{-1}$ | |
| - Aquifer specific yield | $Sy$ | $m^3.m^{-3}$ | |
| - Groundwater recession coefficient | $J$ | $day^{-1}$ | |
| Meteorological forcing | | | van Beek (2008); CRU (Harris et al., 2014); ERA40 (Uppala et al., 2005); ERA-Interim ( Dee et al., 2011) |
| - Total precipitation | $P$ | $m.day^{-1}$ | |
| - Atmospheric air temperature | $T_{air}$ | °C or K | |
| - Reference potential evaporation and transpiration | $E_{ref,pot}$ | $m.day^{-1}$ | |
| Others: | | | |
| - Non-irrigation sectoral water demand (i.e. livestock, dometic and industrial) | | $m.day^{-1}$ | Wada et al (2014) |
| - Desalinated water | | $m.day^{-1}$ | Wada et al., (2011a); FAO (2016) |
| - Lakes and reservoirs | | | GLWD1 (Lehner and Döll, 2004); GranD (Lehner et al., 2011) |

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

**List of Figures**

*Figure 1. Schematic overview of a PCR-GLOBWB 2 cell and its modelled states and fluxes. $S_1$, $S_2$ (soil*
*moisture storage), $S_3$ (groundwater storage), $Q_{dr}$ (surface runoff – from rainfall and snowmelt), $Q_{sf}$ (interflow*
*or stormflow), $Q_{bf}$ (baseflow or groundwater discharge), Inf (riverbed infiltration from to groundwater). The*
*thin red lines indicate surface water withdrawal, the thin blue lines groundwater abstraction, the thin red*
*dashed lines return flows from surface water use and the thin dashed blue lines return flows from groundwater*
*use surface. For each sector: withdrawal - return flow = consumption. Water consumption adds to total*
*evaporation. In the figure, the five modules that make up PCR-GLOBWB 2 is portrayed on the model*
*components.*

*Figure 2. Maps of correlation between simulated and observed discharge time series for 3597 GRDC*
*discharge stations; a. results for the 5 arc-minutes simulation; b. difference between results for 5 arc-minutes*
*and 30 arc-minutes simulation.*

*Figure 3. Histograms of evaluation statistics showing the correlation and Kling-Gupta efficiency (KGE)*
*values for the simulated discharge for the 30 arc-minutes and the 5 arc-minute simulations based on 3597*
*GRDC discharge stations, a. correlation 30 arc-minute simulation, b. correlation 5 arc-minute simulation, c.*
*KGE 30 arc-minute simulation, d. KGE 5 arc-minute simulation, note: the percentage catchments with KGE <*
*-1 are 21% and 12% for 30 and 5 arc-minutes respectively.*

*Figure 4. Cumulative frequency distributions of Kling-Gupta efficiency (KGE) values for GRDC stations that*
*are positioned below (a) and above (b) 1000 m a.m.s.l. It can be expected that for the stations above 1000 m,*
*the upstream area is influenced by snow dynamics.*

*Figure 5. Histograms of evaluation statistics showing the anomaly correlation for the simulated discharge for*
*the 30 arc-minutes and the 5 arc-minute simulations based on 3597 GRDC discharge stations, a. anomaly*
*correlation half arc-degree simulation, b. anomaly correlation 5 arc-minute simulation.*

*Figure 6. Map showing for the 5 arc-minute run the difference between the correlation and the anomaly*
*correlation between simulated and observed discharge time series for 3597 GRDC discharge stations,*
*negative values mean that the correlation is higher than the anomaly correlation.*

*Figure 7. Comparison of PCR-GLOBWB 2 total water storage trends (m/year) with those estimated with*
*GRACE over the period 2003-2015. a. TWS trends simulated with PCR-GLOBWB at 5 arc-minutes resolution*
*(~10 km at the equator). Negative values indicate declining TWS (e.g. groundwater depleted regions). b. TWS*
*trends obtained based on the GRACE JPL PL-RL05M Mascon product. The GRACE data were resampled to*
*the resolution of 30 arc-minutes, but they actually represent the 3 x 3 arc-degree (~300 km x 300 km) area,*
*which is the native resolution of the GRACE signal.*

*Figure 8. a. Correlation between monthly TWS time series simulated PCR-GLOBWB 2 and the GRACE JPL*
*PL-RL05M Mascon product over the period 2003-2015. b. Comparison of annual TWS series (inter-annual*
*variability). Comparison is only done for the larger basins over 900,000 km2, conform the 3x3 arc-degree*
*resolution of GRACE.*

*Fig. 9: Country water withdrawal ($km^3$/year) by source, evaluation of simulations with PCR-GLOBWB 2 with*
*reported values in AQUASTAT (FAO, 2016). The scatterplots on the left (a, c, e) are for the period 1968-*
*1992, while the right ones (b, d, f) are 1993-2015. The uppermost plots (a, b) are for total water withdrawal,*
*the middle ones (c, d) are groundwater withdrawal, and the lowermost charts (e, f) are surface water*
*withdrawal. Regression coefficient based on regression to non-log transformed data with intercept kept zero.*

*Fig. 10: Country water withdrawal (km3/year) by sector, evaluation of simulations with PCR-GLOBWB 2*
*with reported values in AQUASTAT (FAO, 2016). The scatterplots on the left (a, c, e) are for the period 1968-*
*1992, while the right ones (b, d, f) are 1993-2015. The uppermost plots (a, b) are for withdrawal for*
*agricultural purpose, the middle ones (c, d) are industrial withdrawal, and the lowermost charts (e, f) are*
*domestic. Regression coefficient based on regression to non-log transformed data with intercept kept zero.*



**List of Tables**


*Table 1. Global water balance components and human water withdrawal (in $km^3$/year and mm/year) over the period 2000-2015 as obtained from the 30 arc-minute and the 5 arc-minute simulations. The numbers are shown to high significance to show the water balance closure. This does not mean that we pretend to know e.g. global discharge with a $km^3$ accuracy (actual accuracy of the large fluxes is more in the order of $10^3$ $km^3$).*


*Table 2. Groundwater withdrawal and total water withdrawal as compared to other studies (in $km^3$/year)*



*Table A1. List (non-exhaustive) of state and flux variables defined in PCR-GLOBWB.*


*Table A2. List of model inputs and parameters.*


*Table 1 - Global water balance components and human water withdrawal (in km³/year and mm/year) over the period 2000-2015 as obtained from the 30 arc-minute*

*and the 5 arc-minute simulations. The numbers are shown to high significance to show the water balance closure. This does not mean that we pretend to know e.g.*

*global discharge with a km³ accuracy (actual accuracy of the large fluxes is more in the order of $10^3$ km³)*

| | | 30 arc-min | | 5 arc-min | |
|---|---|---|---|---|---|
| | | km³/year | mm/year | km³/year | mm/year |
| Global water balance | Precipitation | 107452 | 808 | 107495 | 811 |
| | Desalinated water use | 3 | 0.02 | 2 | 0.01 |
| | Runoff | 42393 | 319 | 43978 | 332 |
| | Evaporation* | 65754 | 494 | 63974 | 483 |
| | Change in total water storage | -693 | -5 | -455 | -3 |
| Groundwater budget | Groundwater recharge | 27756 | 209 | 25521 | 193 |
| | Groundwater withdrawal | 737 | 6 | 632 | 5 |
| | Non-renewable groundwater withdrawal (groundwater depletion) | 173 | 1 | 171 | 1 |
| | Renewable groundwater withdrawal | 564 | 4 | 460 | 3 |
| Withdrawal by sector | Agricultural water withdrawal (irrigation + livestock) | 2735 | 21 | 2309 | 17 |
| | Domestic water withdrawal | 380 | 3 | 314 | 2 |
| | Industrial water withdrawal | 798 | 6 | 707 | 5 |
| Withdrawal by source | Total water withdrawal | 3912 | 29 | 3330 | 25 |
| | Surface water withdrawal | 3172 | 24 | 2697 | 20 |
| | Desalinated water use | 3 | 0.02 | 2 | 0.01 |
| | Groundwater withdrawal | 737 | 6 | 632 | 5 |

* Includes consumptive water use for livestock, domestic and industrial sectors

*Table 2 - Groundwater withdrawal and total water withdrawal as compared to other studies (in*
*km³/year)*

| | Source | Year | Value (km³/year) |
|---|---|---|---|
| Groundwater withdrawal | Wada et al. (2010) (from the IGRAC database) | 2000 | 734 (±87) |
| | Döll et al. (2012) | 1998-2002 | 571 |
| | Döll et al. (2014) (their Table 2). | 2003-2009 | 690-888 |
| | Döll et al. (2014) (their Table 6). | 2000-2009 | 665 |
| | Pokhrel et al. (2015) | 1998-2002 | 570 (±61) |
| | Hanasaki et al. (2018) | 2000 | 789 (±30) |
| | This study (5 arc-minutes) | 2000-2015 | 632 |
| Total water withdrawal | Vörösmarty et al. (2005) | 1995-2000 | 3560 |
| | Oki and Kanae (2006) | contemporary | 3800 |
| | Döll et al. (2012) | 1998-2002 | 4340 |
| | Döll et al. (2014) (their Table 2) | 2003-2009 | 3000-3700 |
| | FAO (2016) | 2010 | 3583 |
| | Hanasaki et al. (2018) | 2000 | 3628 (±75) |
| | This study (5 arc-minutes) | 2000-2015 | 3330 |


*Table A1 - List (non-exhaustive) of state and flux variables defined in PCR-GLOBWB*

| Description | Symbol | Unit |
|---|---|---|
| Interception storage | $S_{int}$ | m |
| Snow cover/storage in water equivalent thickness (excluding liquid part $S_{slq}$) | $S_{swe}$ | m |
| Liquid/melt water storage in the snow pack | $S_{slq}$ | m |
| Upper and lower soil storages | $S_1$ and $S_2$ | m |
| Surface water storage (lakes, reservoirs, rivers and inundated water) | $S_{wat}$ | m |
| groundwater storage (renewable part) | $S_3$ | m |
| fossil groundwater storage (non-renewable) | $S_{nrw}$ | m |
| total groundwater storage = $S_3 + S_{nrw}$ | $S_{gwt}$ | m |
| total water storage thickness = $S_{int} + S_{swe} + S_{slq} + S_1 + S_2 + S_{gwt}$ | TWS | m |
| | | |
| potential evaporation | $E_{pot}$ | m.day$^{-1}$ |
| evaporation flux from the intercepted precipitation | $E_{int}$ | m.day$^{-1}$ |
| evaporation from melt water stored in the snow pack | $E_{slq}$ | m.day$^{-1}$ |
| bare soil evaporation | $E_{soil}$ | m.day$^{-1}$ |
| transpiration from the upper and lower soil stores | $T_1$ and $T_2$ | m.day$^{-1}$ |
| total land evaporation = $E_{int} + E_{slq} + E_{soil} + T_1 + T_2$ | $E_{land}$ | m.day$^{-1}$ |
| surface water evaporation | $E_{wat}$ | m.day$^{-1}$ |
| total evaporation = $E_{land} + E_{wat}$ | $E_{tot}$ | m.day$^{-1}$ |
| | | |
| direct runoff | $Q_{dr}$ | m.day$^{-1}$ |
| interflow, shallow sub-surface flow | $Q_{sf}$ | m.day$^{-1}$ |
| baseflow, groundwater discharge | $Q_{bf}$ | m.day$^{-1}$ |
| specific runoff from land | $Q_{loc}$ | m.day$^{-1}$ |
| local change in surface water storage | $Q_{wat}$ | m.day$^{-1}$ |
| total specific runoff | $Q_{tot}$ | m.day$^{-1}$ |
| routed channel (surface water) discharge | $Q_{chn}$ | m$^3$.sec$^{-1}$ |
| | | |
| net fluxes from the upper to lower soil stores | $Q_{12}$ | m.day$^{-1}$ |
| net groundwater recharge, fluxes from the lower soil to groundwater stores | RCH = $Q_{23}$ | m.day$^{-1}$ |
| surface water infiltration to groundwater | Inf | m.day$^{-1}$ |
| | | |
| desalinated water withdrawal | $W_{sal}$ | m.day$^{-1}$ |
| surface water withdrawal | $W_{wat}$ | m.day$^{-1}$ |
| renewable groundwater withdrawal | $W_3$ | m.day$^{-1}$ |
| non-renewable groundater withdrawal (groundwater depletion) | $W_{nrw}$ | m.day$^{-1}$ |
| total groundwater withdrawal = $W_3 + W_{nrw}$ | $W_{gwt}$ | m.day$^{-1}$ |


*Table A1 - continued*

| Description | Symbol | Unit |
|---|---|---|
| water withdrawal allocated for irrigation purpose | $A_{irr}$ | m.day$^{-1}$ |
| water withdrawal allocated for livestock demand/sector | $A_{liv}$ | m.day$^{-1}$ |
| water withdrawal allocated for agricultural sector $= A_{irr} + A_{liv}$ | $A_{agr}$ | m.day$^{-1}$ |
| domestic water withdrawal | $A_{dom}$ | m.day$^{-1}$ |
| industrial water withdrawal | $A_{ind}$ | m.day$^{-1}$ |


*Table A2 - List of model inputs and parameters*

| Description | Symbol | Unit | References/sources |
|---|---|---|---|
| Upper and lower soil store parameters: | | | FAO (2007) soil map; van Beek and Bierkens (2009) |
| - Soil thickness | $Z_1$ and $Z_2$ | m | |
| - Residual soil moisture content | $\theta_{r-1}$ and $\theta_{r-2}$ | $m^3.m^{-3}$ | |
| - Soil moisture at saturation | $\theta_{s-1}$ and $\theta_{s-2}$ | $m^3.m^{-3}$ | |
| - Soil water storage capacity per soil layer: $SC = Z / (\theta_s - \theta_r)$ | $SC_1$ and $SC_2$ | m | |
| - Soil matric suctions at saturation | $\psi_{s-1}$ and $\psi_{s-2}$ | m | |
| - Exponent in the soil water retention curve | $\beta_1$ and $\beta_2$ | dimensionless | |
| - Saturated hydraulic conductivities of upper and lower soil stores | $K_1$ and $K_2$ | $m.day^{-1}$ | |
| - Total soil water storage capacities $= SC_{upp} + SC_{low}$ | $W_{max}$ | m | |
| Land cover fraction: Land cover areas (including extent of irrigated areas) over cell areas | $f_{lcov}$ | $m^2.m^{-2}$ | GLCC v2.0 map (USGS, 1997); Olson (1994a, 1994b); MIRCA2000 dataset (Portmann et al., 2010), FAOSTAT (2012) |
| Topographical parameters: | DEM | m | HydroSHEDS (Lehner et al., 2008); Hydro1k (Verdin and Greenlee, 1996); GTOPO30 (Gesch et al., 1999) |
| - Cell-average DEM | $DEM_{avg}$ | m | |
| - Flood plain elevation | $DEM_{fpl}$ | m | |

 *Table A2 - Continued*

| Description | Symbol | Unit | References/sources |
| --- | --- | --- | --- |
| Root fractions per soil layer | $Rf_{upp}$ & $Rf_{low}$ | dimensionless | Canadell et al. (1996); van Beek and Bierkens (2009) |
| Arno scheme (Todini, 1999; Hagemann and Gates, 2003) exponents defining soil water capacity distribution | $\beta_{arno}$ | dimensionless | Canadell et al. (1996), Hagemann et al. (1999); Hagemann (2002); van Beek (2008); van Beek and Bierkens (2009) |
| Ratio of cell-minimum soil storage to $W_{max}$ | $f_{wmin}$ | m | van Beek (2008); van Beek and Bierkens (2009) |
| Ratio of cell-maximum soil storage to $W_{max}$ | $f_{wmax}$ | m | van Beek (2008); van Beek and Bierkens (2009) |
| Parameters related to phenology | | | Hagemann et al. (1999); Hagemann (2002); van Beek (2008); van Beek and Bierkens (2009) |
| - Crop coefficient | $K_c$ | dimensionless | |
| - Interception capacity | $S_{int-max}$ | m | |
| - Vegetation cover fraction | $C_v$ | $m^2.m^{-2}$ | |
| Groundwater parameters | | | GLHYMPS map (Gleeson et al., 2014); van Beek (2008); van Beek and Bierkens (2009) |
| - Aquifer transmissivity | $KD$ | $m^2.day^{-1}$ | |
| - Aquifer specific yield | $Sy$ | $m^3.m^{-3}$ | |
| - Groundwater recession coefficient | $J$ | $day^{-1}$ | |

*Table A2 - Continued*

| Description | Symbol | Unit | References/sources |
|---|---|---|---|
| Meteorological forcing: | | | van Beek (2008); CRU (Harris et al., 2014); ERA40 (Uppala et al., 2005); ERA-Interim ( Dee et al., 2011) |
| - Total precipitation | $P$ | m.day$^{-1}$ | |
| - Atmospheric air temperature | $T_{air}$ | °C or K | |
| - Reference potential evaporation and transpiration | $E_{ref,pot}$ | m.day$^{-1}$ | |
| Others: | | | |
| - Non-irrigation sectoral water demand (i.e. livestock, dometic and industrial) | | m.day$^{-1}$ | Wada et al (2014) |
| - Desalinated water | | m.day$^{-1}$ | Wada et al., (2011a); FAO (2016) |
| - Lakes and reservoirs | | | GLWD1 (Lehner and Döll, 2004);  GranD (Lehner et al., 2011) |
