# Peer review of "PCR-GLOBWB 2: a 5 arc-minute global hydrological and water resources model"

_Geoscientific Model Development, 2017_

## Referee Comment (RC1) · Anonymous Referee #1 · 25 Mar 2018

Review of gmd-2017-288

PCR-GLOBWB 2: a 5 arc-minute global hydrological and water resources model Sutanudjaja et al.

Summary:

――――

This is a well-written manuscript that describes the components of PCR-GLOBWB 2 and its updates since version 1. It also includes an evaluation of a global application of the model at 5 and 30 arc-minute. While no individual component of the model is entirely new (i.e. most of the components have been the subject of their own publications), this manuscript serves a useful purpose in providing a complete overview of the

updated model. As such, I recommend that the manuscript is acceptable for publication in GMD pending minor revisions

Comments:

———

l.68: 'om' should read 'on'

l.73: What does 'global water management' mean? Most water management decisions are made locally or regionally (e.g. by basin). It is not clear to me which water management decisions are made globally.

l.86-88: How many of these publications had someone not closely affiliated with the Utrecht group as their primary authors?

l.88: 'yielded 97 publications with collectively over 2100 references'. I assume you mean citations rather than references (since it would be easy to add more references to a paper).

l.104-106: Is the resolution relevant from a code-perspective? I assume that the code itself is resolution-agnostic and that the resolution of the application is specified in configuration and other input files?

l.119: The experiments described in the manuscript serve to 'evaluate' the model rather than to 'validate' it.

l.178-179: '[...] in which the exchange of water between a series of interconnected stores is easily performed'. Awkward phrasing. You don't 'perform exchange' and it's unclear what 'easily' means in this context.

l.180: Provide more details on the modular construction of the model at the code level. This is of interest to readers of GMD.

Figs 1 and 2: Combine into one single figure.
[Figure]

l.196-210: Provide proper references.

l.239-240: How are rain-fed crops handled?

l.239-240: How easy / hard would it be for users to add additional / different land use types (e.g. urban or tundra)

l.251: Describe how 'Darcian flow' is implemented. Is this vertical drainage only under unit gradient? How is the unsaturated hydraulic conductivity specified or calculated?

l.276-278: Are the 'where under natural conditions (without groundwater withdrawal) significant groundwater discharge occurs' dynamically calculated or specified?

l.287: ': Harbaugh et al., 2000)' should read '(Harbaugh et al., 2000)'

l.301-302: At what resolution is the 8-point steepest gradient algorithm evaluated? I assume at a higher resolution than 5-min or 30-min, because neither will result in accurate channel networks if the steepest gradient algorithm is applied using cell-average elevations at those resolutions.

l.302-303: What happens when river flow is routed in an endorheic basin? Does it create an inland sea or is the water removed from the model?

l.304-313: Fix grammar and punctuation. As is, the enumeration and the associated semi-columns make no sense since there are multiple sentences after '2)'.

l.326: How is reservoir management handled? For example, are releases based on storage targets, rule curves, etc. How are different reservoir purposes addressed, e.g. flood control, hydropower, irrigation.

l.329: What is a 'standard' storage-outflow relationship?

l.335: What is 'water type'?

Section 2.3.4: Are inter-basin water transfers / diversions represented?

Section 2.3.5: It is not clear to me how the irrigation efficiency is used in the model.

I understand it can be used to estimate the water demand, but what happens to the water after it is removed from storage or the river network. Is the excess water (the 'inefficient' portion) added to the soil (where it can then contribute to evapotranspiration and return flow), is it directly added to return flows, or something else?

l.399: 'would be rather straightforward to change this'. Explain in one sentence how that would be done.

l.403: 'as function' should read 'as a function'

l.413: 'to use literature fractions of groundwater withdrawal and surface water withdrawal'. Awkward, suggested change: 'to use fractions of groundwater and surface water withdrawal reported in the literature'.

Section 2.5: Does the model use openmp (shared memory) or mpi parallelization?

l.485: Provide details on the memory constraints.

Section 3.1.1: Since neither model implementation was calibrated it makes it a challenging to evaluate the statements that say one version (5-arcmin) is inherently better than the other one (30-arcmin).

l.600: 'scale-consistent' - perhaps 'scale-independent'?

Table 1 and comparison of 5- and 30-arcmin resolutions: One way to reduce difference that may occur because of area mismatches would be to provide fluxes also as a depth per unit area, e.g. mm/year.

Section 3.4. and following: When providing error metrics (correlation, KGE, etc.) please provide the timestep at which the metric was calculated.

l.543: The term 'hydrological extremes' as used here is a bit misleading. For many of the smaller basins, the time-of-concentration is well less than the timestep used in the evaluation of the error metrics. For example, the monthly flow in a 2000km2 basin is not necessarily related to a flood event.

Section 3.4.2: When comparing with GRACE it makes more sense to scale the simulations to the resolution of GRACE (as is done in the analysis for Figure 8) than comparing the GRACE results directly to the 5- or 30-arcmin results. I suggest removing Figure 7 and the accompanying discussion and to focus on Figure 8 instead.

Figures 9 and 10: Distinguish the left and right columns in the caption and in figure labels.

---

## Referee Comment (RC2) · Anonymous Referee #2 · 10 Apr 2018

Major comments

The authors compiled their earlier modeling efforts and upgraded the PCR-GLOBWB global hydrological model. This paper consists of two parts, model description and validation. The latter part particularly focused on the comparison between two global simulations of 30 arc min and 5 arc min spatial resolutions. The authors claim that the simulation of finer resolution generally outperform the other.

I found the former part well written except for some technical issues listed below. I am concerned by the validity of discussion of the latter part. The authors mainly compared the histogram of several hydrological indicators for two spatial resolutions. This straightforward approach is sometimes misleading because the performance improvement in some specific conditions can be exaggerated. For instance, because elevation

correction is applied for air temperature, the performance of finer resolution is expected better than the other in snow-dominant mountainous regions. Because the river gauging stations are concentrated in northern mid and high latitudes, the effect tends to contrast the performance of two resolutions. Performance improvement must be evaluated with more careful investigations. Second, the performance of water use estimation is questionable. The results indicate that estimated national water use differs from AQUASTAT by one or two orders of magnitude. Since little discussion is provided to these considerable discrepancies, I'm puzzled how I should take these results. Further clarification and reasonable discussion should be added to the water use section.

Specific comments

Line 52 "H08 (Hanasaki et al 2008a)": H08 seems recently updated (Hanasaki et al. 2018). The paper may be of interest of the authors because some of the model functions are overlapping with PCR-GLOBWB 2.

Line 229 "resulting crop specific potential evaporation": Do the authors estimate potential evaporation of trees as well? If this is the case, "vegetation specific" may look better.

Line 239 "tall natural vegetation, short natural vegetation, irrigated crops and paddy-irrigation": How are rainfed crops treated in this model?

Line 245 "using a monthly climatology of phenology and crop calendars": If the crop calendars are monthly, crops are always planted at the first day of month and harvested at the last day. Is this the case of this model?

Line 256 "All fluxes are computed per land cover type and balanced with the available storage to arrive...": Are storage terms computed independently for each land cover type? For instance, is the soil moisture of natural vegetation different from that of irrigated cropland? If not, how water is balanced with the available storage?

Line 284 "Alternatively, an initial estimate of a fossil, i.e. a non-actively replenished,

groundwater store can be imposed that provides a similar functionality": Hard to read. Rephrase.

Line 365 "the crop composition (which changes per month and includes multi-cropping)": Same question as above. If the crop calendar used is monthly, does it mean all the crops are planted and harvested at the first and the last day of month globally?

Line 378 "the irrigation water demand is increased by 40% to obtain gross irrigation water demand": Is there any rationale for this coefficient? In Section 3.4.3, the authors simply attributed the underestimated irrigation water withdrawal to this coefficient.

Line 383 "the gross demand and net demand are prescribed to the model and calculated using separate script": Confusing. Are gross and net demand prescribed or calculated?

Line 445 "2.4 Differences between PRC-GLBWB 1 and 2": This section seems better to be placed after "2.5 Model code".

Line 469 "tailor-made built-in hydrological function": Hard to read. What does it mean?

Line 470 "its syntax that reads like pseudo-code, generally results in short and readable model codes...": Sounds a bit subjective. Describe objective characteristic of code.

Line 505 "Note that parameterizations were derived directly following their source data sets using hydrological concepts described in Van Beek and Bierkens (2009)": "The way of setting hydrological parameters are unchanged from Van Beek and Bierkens (2009)"? Is this what the authors meant here? Anyway, it includes little useful information. On what parameterizations do the authors discussing here?

Line 515 "We used ERA40 and ERA-I results that had been resampled by ECMWFs resampling scheme from their original resolutions to 30 arc-minutes": What do you mean by resampling? Is this different from spatial interpolation? Elaborate methodology and some reasons for adopting the technique.

Line 522 "Equally monthly reference potential evaporation, computed with Penman-Monteith from the CRU data set was...downscaled to daily data proportional to Hamon evaporation...": It would be better to state the background or key reasons for this procedure. Why didn't you solve the Penman-Monteith equation and directly derive daily potential evaporation by using ERA40 or ERA-Interim?

Line 562 "main river in PCR-GLOBWB": What is the main river?

Line 563 "This yielded 5363 stations for the 5 arc-minute simulation, 3910 stations for the 30 arc-minute simulation": I'm interested in the distribution of catchment area of these stations. For instance, the number of station for 30 arc-minute is smaller than 5 arc-minute one. Is this mainly because the stations below ∼2500 km2 of catchment area (an approximate area of a single grid cell) cannot be represented by 30 arc-minute? To answer such questions, why don't you show the maximum, minimum, mean, and median of catchment area for each spatial resolution?

Line 602 "regionalization": What is this? Do you mean optimization (or tuning) of hydrological parameters to reproduce historical records?

Line 603 "scale-consistent flux-preserving": Hard to know what is meant here. Rephrase

Line 604 "parameterization is possible": What kind of parameterization is mentioned here? What does possible mean?

Line 623 Table 1: It is interesting to compare with earlier works (e.g. Table 2 and 5 of Hanasaki et al. 2018).

Line 635 "cross-correlation": Which did you use cross-correlation (a technique frequently used in signal processing) or Pearson's correlation? I'm asking this because the results indicate the authors used Pearson's correlation, but they always wrote cross-correlation throughout the text.

Line 647 "Figure 3": It is hard to see the differences between a (30min) and b (5min).

[Figure]

Why don't you show the difference between two?

Line 648 "Figure 4": As Figure 3 clearly indicates, the selected GRDC stations are concentrated in Europe. Figure 4 might be a bit misleading if the majority of stations are concentrated in some specific regions (e.g. snow dominated stations in Europe). My suggestion is to make Figure 4 for major climatic zones. It would be useful to identify in which climatic conditions the results are improved. As mentioned above, it would be also interesting to separate Figure 4 by catchment area. I suspect the improvements are concentrated in relatively small basins. Another point is that total frequency apparently far exceeds the number of stations (3597). Elaborate how to see these panels (same for Figure 5).

Line 694 "in case of the Niger River, not representing the inner delta...": Or simply something wrong with input or validation data.

Line 653 "a better delineation of the outline of the basins...": You mentioned that the error in catchment area is less than 15% for all basins for either spatial resolutions. Further elaborate what do you mean by "better delineation of the outline" here.

Line656 "better snow dynamics due to the downscaling of temperature to 5 arc-minute resolution": Similar comment to above. This argument must be easily supported by showing the performance of snow dominated regions for two simulations (i.e. excluding snow free regions from Figure 4).

Line 670 "Although results are generally better, the spatial distribution of results is similar to those found by Van Beek et al. (2011) for PCR-GLOBWB1". This conveys hardly any information. What does "generally better" mean? What are similar and what are not?

Line 688 "indicating a higher skill with regard to capturing extremes and anomalies": I'm not convinced at all. As mentioned above, the performance must be different by catchment area, climate, topography and other factors. Show concrete evidences for

this claim.

Line 763 "Also, Figure 10 shows that agricultural water withdrawal is underestimated . . .": I'm quite puzzled by the right panel of Figure 10a. First, majority of plots are located far below the y=x line indicating most countries are underestimated one (or two) order of magnitude. Next, although majority of countries are strongly underestimated, the correlation slope is larger than 1 which indicates the overall results are overestimated due to some outliers' behavior. These points should be more highlighted to call readers' attention. Finally, my honest interpretation of Figures 9 and 10 is that this model fails to reproduce the historical dynamics of country-specific water withdrawal. At best, the simulation outputs are considerably different from AQUASTAT. Further clarify the authors' intention to show Figures 9 and 10 together with discussion on the capability and limitation of the water use module of PCR-GLOBWB 2.

Line 786 "Simulated water withdrawal, by source and sector, matches reasonably well with reported water withdrawal from AQUASTAT": I'm not able to agree with this statement. The authors reported that the regression slope was as low as 0.54 for some cases (line 761).

Technical comments

Line 67 "Schewe et al. 2013; Haddeland et al. 2013": Check publication year of these articles. It must be 2014.

Line 89 "collectively over 2100 references": do you mean citations?

References

Haddeland, I., Heinke, J., Biemans, H., Eisner, S., Flörke, M., Hanasaki, N., Konzmann, M., Ludwig, F., Masaki, Y., Schewe, J., Stacke, T., Tessler, Z. D., Wada, Y., and Wisser, D.: Global water resources affected by human interventions and climate change, P. Natl. Acad. Sci. USA, 111, 3251-3256, 10.1073/pnas.1222475110, 2014.

Hanasaki, N., Yoshikawa, S., Pokhrel, Y., and Kanae, S.: A global hydrological simulation to specify the sources of water used by humans, Hydrol. Earth Syst. Sci., 22, 789-817, 10.5194/hess-22-789-2018, 2018.

Schewe, J., Heinke, J., Gerten, D., Haddeland, I., Arnell, N. W., Clark, D. B., Dankers, R., Eisner, S., Fekete, B. M., Colón-González, F. J., Gosling, S. N., Kim, H., Liu, X., Masaki, Y., Portmann, F. T., Satoh, Y., Stacke, T., Tang, Q., Wada, Y., Wisser, D., Albrecht, T., Frieler, K., Piontek, F., Warszawski, L., and Kabat, P.: Multimodel assessment of water scarcity under climate change, P. Natl. Acad. Sci. USA, 111, 3245-3250, 10.1073/pnas.1222460110, 2014.

---

## Author Comment (AC1) · 12 May 2018

We would like to thank both referees for their constructive comments. Please find our responses in a form of supplement as a zip file. The zip file contains two pdf files. The first one, "replies_and_revised_paper_with_track_changes.pdf", contains our replies for both referees and the revised manuscript document that includes track changes. The second file, "revised_paper.pdf", is the revised manuscript document but without track changes.

Please also note the supplement to this comment:
https://www.geosci-model-dev-discuss.net/gmd-2017-288/gmd-2017-288-AC1-
supplement.zip

<printer-friendly version>

---

## Author Response (AR1)

**Response to Referee #1's review on the GMD-D paper of Sutanudjaja et al. (2017):**

Note: *The referee comments are given in blue italics.* Our answers are in black. The "Response to Referee #2" is also given in this document starting from the page 12. The revised manuscript with track changes is given after the "Response to Referee #2" (after page 25). Please also be understood that all line numbers mentioned in our answers are referring to the revised manuscript without track changes (given in another file).

*Review of gmd-2017-288*

*PCR-GLOBWB 2: a 5 arc-minute global hydrological and water resources model Sutanudjaja et al.*

*Summary:*

*———*

*This is a well-written manuscript that describes the components of PCR-GLOBWB 2 and its updates since version 1. It also includes an evaluation of a global application of the model at 5 and 30 arc-minute. While no individual component of the model is entirely new (i.e. most of the components have been the subject of their own publications), this manuscript serves a useful purpose in providing a complete overview of the updated model. As such, I recommend that the manuscript is acceptable for publication in GMD pending minor revisions.*

We would like to thank Referee #1 for her/his kind appraisal for our paper and valuable suggestions and comments. We concur the importance of this paper to provide a formal and complete overview of our PCR-GLOBWB 2 model that has been used in many studies.

*Comments:*

*———*

*l.68: 'om' should read 'on'*

Thank you for pointing this. We have corrected it in the revised paper.

*l.73: What does 'global water management' mean? Most water management decisions are made locally or regionally (e.g. by basin). It is not clear to me which water management decisions are made globally.*

We rephrased the sentence to: "These applications show that GHMs have become invaluable tools in support of global change research and environmental assessments."

*l.86-88: How many of these publications had someone not closely affiliated with the Utrecht group as their primary authors?*

A recent search on Scopus (13 April 2018) using the key-word "PCR-GLOBWB" yielded 113 publications with collectively over 2500 citations. There are 50 publications (44%) without authors from our group (i.e. by excluding "Utrecht University" during the search on Scopus).

*l.88: 'yielded 97 publications with collectively over 2100 references'. I assume you mean citations rather than references (since it would be easy to add more references to a paper).*

We modified it accordingly.

*l.104-106: Is the resolution relevant from a code-perspective? I assume that the code itself is resolution-agnostic and that the resolution of the application is specified in configuration and other input files?*

Indeed, the reviewer is right. The resolution is specified in configuration and other input files. We rephrase the sentence to the following (see also lines 104-106 of the revised manuscript without track changes).

"The new version of the model, PCR-GLOBWB 2, which is able to simulate the water balance at a finer spatial resolution of 5 arc-minutes, supersedes the original PCR-GLOBWB 1, which has a resolution of 30 arc-minutes only."

*l.119: The experiments described in the manuscript serve to 'evaluate' the model rather than to 'validate' it.*

We modified it accordingly. In the revised manuscript, the terms "validation", "validate", and "validated" were not used. They have been replaced replaced with "evaluation", "evaluate", and "evaluated" (see e.g. lines 117-119 of the revised manuscript).

*l.178-179: '[...] in which the exchange of water between a series of interconnected stores is easily performed'. Awkward phrasing. You don't 'perform exchange' and it's unclear what 'easily' means in this context.*

We removed the phrase.

*l.180: Provide more details on the modular construction of the model at the code level. This is of interest to readers of GMD.*

We appreciate the suggestion. However, we think that such details (at the code level) are not beneficial for the manuscript and general readers as they are hardly presented in a concise and straightforward manner. For readers that are interested with the model code, we refer to the PCR-GLOBWB github page [https://github.com/UU-Hydro/PCR-GLOBWB_model](https://github.com/UU-Hydro/PCR-GLOBWB_model) (in the manuscript, see also Section 5 "Code and data availability").

Nevertheless, regarding to this comment, we add the following sentences to the revised manuscript in order to briefly state our general approach in developing the model code of PCR-GLOBWB 2 (see Section 2.4, lines 48-453 of the revised manuscript without track changes).

"To allow for exchanges of model components and, therefore, evaluate different model configurations, a component-based development approach (e.g Argent, 2004; Castronova and Goodall, 2010) was followed while developing the PCR-GLOBWB 2 model code. Each of the PCR-GLOBWB scientific modules described in section 2.3 is implemented in a separate Python class that needs to implement initialization and update methods. The latter designates changes of states and fluxes per time step. Each of module is initialized and executed by iteratively calling the update method via a main model script."

*Figs 1 and 2: Combine into one single figure.*

We modified it accordingly.

*l.196-210: Provide proper references.*

As far as we are concerned, the proper references were already provided. In the revised manuscript, we tried to improve them by putting the link inside the brackets for each reference. This may avoid unnecessary confusion. Please kindly let us know if there are more things that we should do (see lines 182-197).

*l.239-240: How are rain-fed crops handled?*

The PCR-GLOBWB land cover classes used for a demonstration in this manuscript consist of four land cover classes: 'tall natural vegetation', 'short natural vegetation', 'non-paddy irrigated crops', and 'paddy-irrigation'. Here we simplified that the "rain-fed crops" was merged to the 'short natural vegetation'. Nevertheless, there is also a version (Bosmans et al., 2017) that consists of six land cover classes and handles the "rain-fed crops" in a separate class, albeit still at 30 arc-minute resolution only.

Related to this comment, we have revised the first paragraph of Section 2.3.2 as follows:

"This core module of PCR-GLOBWB 2 covers the land-atmosphere exchange, the vertical flow between soil compartments and the eventual groundwater recharge, snow and interception storage and the runoff generation mechanisms. These processes are simulated over a number of land cover types and aggregated proportionally based on land cover fractions within a model cell. Users can specify their own land cover classification and introduce their own land cover parameterization. The standard parameterization of PCR-GLOBWB 2 carries four land cover types consisting of tall natural vegetation, short natural vegetation, non-paddy irrigated crops, and paddy irrigated crops (i.e. wet rice). There is also a parameterization set for six land cover types (Bosmans et al., 2017), albeit still at 30 arc minute resolution only, that include distinct types for pasture and rain-fed crops. For the standard four land cover parameterization of PCR-GLOBWB, applied in this paper, the land cover types of pasture and rain-fed crops are integrated into the short natural vegetation type."

*l.239-240: How easy / hard would it be for users to add additional / different land use types (e.g. urban or tundra)*

Users can specify their own land cover classification and introduce their own land cover parameterization (as done by Bosmans et al., 2017). Within our department, although not for global extent simulation, there are also some ongoing studies (e.g. master and phd projects) implementing PCR-GLOBWB 2 with their own customized land cover classes (e.g. separating sugar cane in a separate land cover class in order to assess hydrological impacts of its expansion due to growing bio fuel needs).

*l.251: Describe how 'Darcian flow' is implemented. Is this vertical drainage only under unit gradient? How is the unsaturated hydraulic conductivity specified or calculated?*

Net vertical fluxes between the stores 1 and 2 are driven by degrees of saturation of both layers, s. They are calculated either as $s_1 = S_1/SC_1$ and $s_2 = S_2/SC_2$; or $s_1 = \theta_1/\theta_{sat,1}$ and $s_2 = \theta_2/\theta_{sat,2}$, where S is the storage, SC is the storage capacity, $\theta$ is the effective moisture content defined as the fraction of storage over thickness, and the subscript sat indicates saturation. In principle, net vertical fluxes between both stores, $Q_{12}$ consists of a downward percolation $Q_{1\rightarrow2}$ and a capillary rise $Q_{2\rightarrow1}$. If there is enough water in $S_1$, percolation $Q_{1\rightarrow2}$ is equal to the first store unsaturated hydraulic conductivity, $K_1(s_1)$. If $s_1 < s_2$, capillary rise may occur with the amount of $Q_{2\rightarrow1} = K_2(s_2) \times (1-s_1)$, where $K_2(s_2)$ is the second store unsaturated hydraulic conductivity and $(1-s_1)$ is the moisture deficit in the first store. The unsaturated hydraulic conductivity of each layer, K(s), which depends on the degree of saturation s, is calculated based on the relationship suggested by Campbell (1974): $K(s) = Ksat \times s^{2\beta+3}$ where $\beta$ is a soil water retention curve parameter based on the model of Clapp and Hornberger (1978).

Net vertical fluxes between the second and groundwater stores, $Q_{23}$, also consist of a downward percolation $Q_{2\rightarrow3}$ and a capillary rise $Q_{3\rightarrow2}$. Basically, the fluxes $Q_{12}$ and $Q_{23}$ are calculated in a similar fashion as $Q_{1\rightarrow2}$ and $Q_{2\rightarrow1}$ (described above). Yet, the capillary rise $Q_{2\rightarrow3}$ (from the groundwater store) only occurs in areas with shallow groundwater tables and with a condition that the resulting moisture content in the second layer cannot exceed its field capacity.

In the revised manuscript, we do not want to add this lengthy explanation. Rather, we will put just some references as follows (see lines 242-244).

"In the soil column, vertical fluxes are based on driven by degrees of saturation of soil layers and interact with the underlying groundwater store, S3 (see e.g. van Beek and Bierkens, 2009; Sutanudjaja et al., 2011; Sutanudjaja, 2012 for detailed explanation)."

*l.276-278: Are the 'where under natural conditions (without groundwater withdrawal) significant groundwater discharge occurs' dynamically calculated or specified?*

This should be specified before executing a (non-naturalized) model run (e.g. by performing a naturalized condition run beforehand). Yet, this setting (to limit river bed infiltration only in areas 'where under natural conditions' (without groundwater withdrawal) significant groundwater discharge occurs) is actually optional in PCR-GLOBWB 2. It was not used in the current demonstration of PCR-GLOBWB 2. Therefore, to avoid confusion, we decided to remove this phrase.

*l.287: ': Harbaugh et al., 2000)' should read '(Harbaugh et al., 2000)'*

We modified it accordingly.

*l.301-302: At what resolution is the 8-point steepest gradient algorithm evaluated? I assume at a higher resolution than 5-min or 30-min, because neither will result in accurate channel networks if the steepest gradient algorithm is applied using cell-average elevations at those resolutions.*

Indeed, we derived our local drainage map based on a high resolution 30 arc-sec digital elevation model derived by combining the 30 arc-sec HydroSHEDS (Lehner et al., 2008), the 30 arc-sec GTOPO30 (Gesch et al., 1999) and the 1 km resolution of Hydro1k (Verdin and Greenlee, 1996; USGS EROS Data Center, 2006). Please refer to the lines 341-347 of the revised manuscript.

*l.302-303: What happens when river flow is routed in an endorheic basin? Does it create an inland sea or is the water removed from the model?*

We simplified that water flowing to endorheic basins is removed (e.g. via evaporation). Please refer to the lines 292-294 of the revised manuscript.

*l.304-313: Fix grammar and punctuation. As is, the enumeration and the associated semi-columns make no sense since there are multiple sentences after '2)'.*

To fix grammar and punctuation, we modified this part. Please see the first and second paragraphs of Section 2.3.4, particularly the lines 289-310 of the revised manuscript.

*l.326: How is reservoir management handled? For example, are releases based on storage targets, rule curves, etc. How are different reservoir purposes addressed, e.g. flood control, hydropower, irrigation.*

For the runs demonstrated in the manuscript, we just used a simple and globally uniform reservoir management rule. Reservoir releases are estimated as a function of reservoir relative minimum and maximum reservoir capacities (minResvrFrac and maxResvrFrac) and long term (5 year) average discharge (see e.g. Wada et al., 2014, particularly their Equation 11). The default values, used in our runs, for these limits are globally uniform set to 10% and 75% of reservoir capacities. Yet, users can also set their own customized minResvrFrac and maxResvr that can be spatially and temporarily varying and dependent on reservoir purposes (e.g. hydropower reservoirs may have a higher upper limit up to 90%).

*l.329: What is a 'standard' storage-outflow relationship?*

What we meant by 'a standard' storage-outflow relationship is that lake outflow is calculated in analogy to the simple weir formula as the discharge over a rectangular cross section (Bos, 1989).

To address these two comments (l.326 and l.329), we added the aforementioned references in the revised manuscript.

*l.335: What is 'water type'?*

What me meant by 'water type' is related to the classification surface water areas to several water types: river channels, inundated floodplains, lakes and reservoirs. To avoid the confusion, we rephrased the sentence as follows (see lines 325-328 in the revised manuscript).

"All surface water areas, which can be classified into several water types: river channels, inundated floodplains, lakes and reservoirs, are subject to open water evaporation calculated from reference potential evaporation multiplied with factors depending on water types and depths."

*Section 2.3.4: Are inter-basin water transfers / diversions represented?*

Water diversions and inter-basin transfers are limited to the pre-described 0.5-1 arc degree water allocation zones/service areas. Please refer to the lines 415-422.

*Section 2.3.5: It is not clear to me how the irrigation efficiency is used in the model. I understand it can be used to estimate the water demand, but what happens to the water after it is removed from storage or the river network. Is the excess water (the 'inefficient' portion) added to the soil (where it can then contribute to evapotranspiration and return flow), is it directly added to return flows, or something else?*

Indeed, the irrigated water is added to the soil and the hydrological conceptualization of the PCR-GLOBWB land surface module (e.g. the improved Arno scheme of Hagemann and Gates, 2003) to determine how much of this water contributes to evaporation and transpiration, direct runoff, interflow and groundwater recharge/baseflow. By applying this, we realize that the simulated irrigation consumption values (withdrawal – return flow) on a day-by-day basis would not be necessarily consistent to the irrigation efficiency values (usually at the country scale), a priori set in the model input, buy they will be approximately similar at longer time scales.

*l.399: 'would be rather straightforward to change this'. Explain in one sentence how that would be done.*

We removed this phrase. Actually, this is still under development. Yet, although the implementation of this feature may be straightforward, we acknowledge further tests are still needed.

*l.403: 'as function' should read 'as a function'*

We modified it accordingly.

*l.413: 'to use literature fractions of groundwater withdrawal and surface water withdrawal'. Awkward, suggested change: 'to use fractions of groundwater and surface water withdrawal reported in the literature'.*

We modified it accordingly.

*Section 2.5: Does the model use openmp (shared memory) or mpi parallelization?*

No. There is still no openmp/mpi technique used. This is still on our wish list for future development.

*l.485: Provide details on the memory constraints.*

We removed this part (about 'Windows memory constraints') as this may not be entirely true and still require some investigation. We acknowledge that we have little experiences for running PCR-GLOBWB under a Windows operational system. All PCR-GLOBWB 2 developments are done under Linux (including their tests). Previously, before we submitted the manuscript, we received reports from some users that they were not able to run PCR-GLOBWB 2 in their Windows laptop. Yet, recently, about a month ago, one of our partners reported that he managed to run PCR-GLOBWB 2 in his Windows laptop.

*Section 3.1.1: Since neither model implementation was calibrated it makes it a challenging to evaluate the statements that say one version (5-arcmin) is inherently better than the other one (30-arcmin).*

We respectfully beg to differ. We have two versions/resolutions (5 arc-minute and 30 arc-minute) of the model with were parameterized similarly in terms of land-cover specific parameters (soil and vegetation properties) and sub-grid parameterization of surface runoff, interflow and groundwater discharge using the same underlying high resolution (30 arc-sec/1 km datasets). Differences are the resolution of topography, catchment outline and drainage network, but they are again obtained from the same underlying dataset (e.g. HydroSHEDS elevation map, GLCC land cover map, GLHYMPS hydrogeological map). So the models are entirely comparable, apart from their resolution. If anything, calibrating the models would actually hide some of the differences between the resolutions because of model fitting that would lead to hiding both resolution errors and conceptual errors in the fitted parameters and would make extrapolation of the results to poorly gauged regions of the world questionable. So we believe that the models can be compared without calibration. Results show that the 5 arc-minutes model performs better. We provide possible reasons for this in the paper, and, in response to the second reviewer, we analyze the reasons behind this improvement a bit further in the revised version, showing that a better representation of the vertical temperature distribution improves performance (this is not intrinsic to PCR-GLOBWB though) and also a better performance in smaller catchments.

*l.600: 'scale-consistent' - perhaps 'scale-independent'?*

We would like to keep on using the term scale-consistent. Scale-independent pertains to parameters that remain unchanged across scales, while scale-consistent allows parameters to change across scales, but the parameters are derived from basic information in the same manner and they are such that if the fluxes from the finer scale model are aggregated to the scale of the coarser scale model, they are the same as the fluxes directly calculated with the coarser scale model. Scale-consistent thus refers to "representative parameters".

*Table 1 and comparison of 5- and 30-arcmin resolutions: One way to reduce difference that may occur because of area mismatches would be to provide fluxes also as a depth per unit area, e.g. mm/year.*

We added the values in mm/year (see Table 1).

*Section 3.4. and following: When providing error metrics (correlation, KGE, etc.) please provide the timestep at which the metric was calculated.*

We revised the manuscript accordingly; see e.g. the first paragraph of Section 3.4.1.

*l.643: The term 'hydrological extremes' as used here is a bit misleading. For many of the smaller basins, the time-of-concentration is well less than the timestep used in the evaluation of the error metrics. For example, the monthly flow in a 2000km2 basin is not necessarily related to a flood event.*

We agree with the reviewer. We have changed it to the following (see lines 634-637): "This is to test if the model is able to capture the monthly scale and inter-annual anomalies in discharge (i.e. on the monthly scale) when the dominant seasonal trend is removed from observations and simulations".

*Section 3.4.2: When comparing with GRACE it makes more sense to scale the simulations to the resolution of GRACE (as is done in the analysis for Figure 8) than comparing the GRACE results directly to the 5- or 30-arcmin results. I suggest removing Figure 7 and the accompanying discussion and to focus on Figure 8 instead.*

We partly agree with the suggestion to remove the figure and we would like to respectfully request to keep Figure 7. Although we cannot really compare the absolute trend values of PCR-GLOBWB and GRACE (due to different resolutions), this figure is still particularly important to show the groundwater depleted regions based on PCR-GLOBWB simulation and to check their consistencies with GRACE signals.

*Figures 9 and 10: Distinguish the left and right columns in the caption and in figure labels.*

We modified them accordingly.

Note: *The referee comments are given in blue italics.* Our answers are in black. The revised manuscript with track changes is given after the "Response to Referee #2". All line numbers mentioned in our answers are referring to the revised manuscript without track changes (given in another file).

*Major comments:*

*The authors compiled their earlier modeling efforts and upgraded the PCR-GLOBWB global hydrological model. This paper consists of two parts, model description and validation. The latter part particularly focused on the comparison between two global simulations of 30 arc min and 5 arc min spatial resolutions. The authors claim that the simulation of finer resolution generally outperforms the other.*

*I found the former part well written except for some technical issues listed below. I am concerned by the validity of discussion of the latter part. The authors mainly compared the histogram of several hydrological indicators for two spatial resolutions. This straightforward approach is sometimes misleading because the performance improvement in some specific conditions can be exaggerated. For instance, because elevation correction is applied for air temperature, the performance of finer resolution is expected better than the other in snow-dominant mountainous regions. Because the river gauging stations are concentrated in northern mid and high latitudes, the effect tends to contrast the performance of two resolutions. Performance improvement must be evaluated with more careful investigations.*

We thank the reviewer for his/her thoughtful comments about the validation with streamflow data. We have taken this comment to heart and looked more carefully into the explanation of the improvement in evaluation statistics between 30 and 5 arc-minute model versions. We have now additionally provided validation statistics for different Köppen-Geiger Climate zones and for GRDC stations with different altitudes. Results are shown hereafter with the specific comments.

*Second, the performance of water use estimation is questionable. The results indicate that estimated national water use differs from AQUASTAT by one or two orders of magnitude. Since little discussion is provided to these considerable discrepancies, I'm puzzled how I should take these results. Further clarification and reasonable discussion should be added to the water use section.*

We agree with the reviewer that we could have been a bit more critical when discussing the validation results of the water withdrawal data. Upon his/her suggestion we have now added discussion better scrutinizing our results and providing additional explanation of the mismatches. We also suggest what could be improved in the "future work" section. We refer to the specific comments for results on this.

*Specific comments*

*Line 52 "H08 (Hanasaki et al 2008a)": H08 seems recently updated (Hanasaki et al. 2018). The paper may be of interest of the authors because some of the model functions are overlapping with PCR-GLOBWB 2.*

Good suggestion. We have added it to the list (see the line 54 of the revised manuscript without track changes).

*Line 229 "resulting crop specific potential evaporation": Do the authors estimate potential evaporation of trees as well? If this is the case, "vegetation specific" may look better.*

We modified the phrase to: "… resulting land cover specific potential evaporation" (see lines 213-216).

*Line 239 "tall natural vegetation, short natural vegetation, irrigated crops and paddyirrigation": How are rainfed crops treated in this model?*

See our answer for Referee #1 (regarding l. 239-240).

*Line 245 "using a monthly climatology of phenology and crop calendars": If the crop calendars are monthly, crops are always planted at the first day of month and harvested at the last day. Is this the case of this model?*

Yes, this is indeed correct. We use a cropping cycle and phenology that is the same for every year, i.e. a climatology based on long-term average temperature and precipitation cycles (see van Beek, 2008; van Beek and Bierkens, 2009 for details). Obviously, phenology and crop cycles depend on climatic conditions, but accounting for that means including a dynamic vegetation and crop growth model. This is certainly on our wish list for future development.

*Line 256 "All fluxes are computed per land cover type and balanced with the available storage to arrive. . .": Are storage terms computed independently for each land cover type? For instance, is the soil moisture of natural vegetation different from that of irrigated cropland? If not, how water is balanced with the available storage?*

Yes, this is indeed the case. Storage terms are calculated and carried through the simulation per land cover type and only when reported averaged over the cells.

*Line 284 "Alternatively, an initial estimate of a fossil, i.e. a non-actively replenished, groundwater store can be imposed that provides a similar functionality": Hard to read. Rephrase.*

Replaced by: "As an alternative, it is also possible to limit the maximum volume of non-renewable groundwater that can be extracted." (see lines 274-275 in the revised manuscript without track changes).

*Line 365 "the crop composition (which changes per month and includes multicropping)": Same question as above. If the crop calendar used is monthly, does it mean all the crops are planted and harvested at the first and the last day of month globally?*

Yes, this is the case. See the answer to the question regarding Line 245.

*Line 378 "the irrigation water demand is increased by 40% to obtain gross irrigation water demand": Is there any rationale for this coefficient? In Section 3.4.3, the authors simply attributed the underestimated irrigation water withdrawal to this coefficient*

Thank you for pointing this out. We found out that this statement was in error. We used country-specific irrigation efficiency values following Rohwer et al. (2007).

We change this sentence to (see lines 369-371):

"The net irrigation demand is augmented to account for limited irrigation efficiency and losses. In PCR-GLOBWB to obtain irrigation water demand including losses, i.e. gross irrigation demand, net irrigation water demand is multiplied with $(1 + f_l)$, with $f_l$ a country-specific loss factor obtained from Rohwer et al. (2007)."

*Line 383 "the gross demand and net demand are prescribed to the model and calculated using separate script": Confusing. Are gross and net demand prescribed or calculated?*

Both are prescribed. We have removed the part "using separate scripts" as this may be confusing.

*Line 445 "2.4 Differences between PRC-GLBWB 1 and 2": This section seems better to be placed after "2.5 Model code".*

We have moved the section as suggested.

*Line 469 "tailor-made built-in hydrological function": Hard to read. What does it mean?*

We changed "tailor-made" by "pre-existing" (see lines 439-441).

*Line 470 "its syntax that reads like pseudo-code, generally results in short and readable model codes. . .":
Sounds a bit subjective. Describe objective characteristic of code.*

We have change "short and readable" with the more objective term "concise" (see lines 439-441).

*Line 505 "Note that parameterizations were derived directly following their source data sets using
hydrological concepts described in Van Beek and Bierkens (2009)": "The way of setting hydrological
parameters are unchanged from Van Beek and Bierkens (2009)"? Is this what the authors meant here?
Anyway, it includes little useful information. On what parameterizations do the authors discussing here?*

We agree that this may be confusing. We have changed this part to (see lines 499-502):

"The parameterization was mostly unchanged from that given in van Beek and Bierkens (2009), but
newer datasets were used if available, such as the GRAND (Lehner et al., 2011) dataset for reservoirs
and MIRCA (Portmann et al., 2010) for crop areas."

*Line 515 "We used ERA40 and ERA-I results that had been resampled by ECMWFs resampling scheme
from their original resolutions to 30 arc-minutes": What do you mean by resampling? Is this different
from spatial interpolation? Elaborate methodology and some reasons for adopting the technique.*

Resampling means in fact a downscaling technique whereby the values of the larger cells are assigned to
the cell centers and then spatially interpolated using inverse distance interpolation. We changed the
sentence to (see lines 510-513).

"We used ERA40 and ERA-I results that had been resampled by ECMWFs resampling scheme from their
original resolutions (~1.2º and ~0.7º) to 30 arc-minutes. Here, resampling means a form of spatial
downscaling whereby the values of the larger ERA40 and ERA-I grid cells are assigned to the cell centers
and then spatially interpolated onto 30 arc-minute grids."

*Line 522 "Equally monthly reference potential evaporation, computed with Penman-Monteith from the CRU data set was. . .downscaled to daily data proportional to Hamon evaporation. . .": It would be better to state the background or key reasons for this procedure. Why didn't you solve the Penman-Monteith equation and directly derive daily potential evaporation by using ERA40 or ERA-Interim?*

We have added the sentence (see lines 519-521):

"We elected not to calculate Penman-Monteith reference evaporation directly from the ERA40 and ERA-I data, in order to avoid the large calculation times needed to process the required meteorological values".

By this procedure we only need precipitation and and temperature as daily values from ECMWF and the monthly CRU values.

*Line 562 "main river in PCR-GLOBWB": What is the main river?*

We have removed "main".

*Line 563 "This yielded 5363 stations for the 5 arc-minute simulation, 3910 stations for the 30 arc-minute simulation": I'm interested in the distribution of catchment area of these stations. For instance, the number of station for 30 arc-minute is smaller than 5 arc-minute one. Is this mainly because the stations below ~2500 km2 of catchment area (an approximate area of a single grid cell) cannot be represented by 30 arcminute? To answer such questions, why don't you show the maximum, minimum, mean, and median of catchment area for each spatial resolution?*

The reason is not the minimum size of 2500 km2, but the first criterion: "allowing a not more than 15% difference in catchment area between PCR-GLOBWB 2 and the area reported with the GRDC discharge station". Because catchments sizes differ between resolutions, this criterion results in different results. It is a good suggestion to provide the catchments sizes. We now mention these numbers in the text (see lines 560-562):

arc-minutes: min=28.2km2 median=2729.9km2 max=4.68e+6km2
arc-minutes: min=31.0km2 median=6560.0km2 max=4.68e+6km2

The small minimum size for the 30 arc-minutes resolution seems to be at odds at first sight. However, we have a Lat-Lon grid, which makes even the 30 arc-seconds cells very small for high latitudes. So, small catchments in high latitudes (mostly the Arctic) can be resolved with 30 arc-minute cells.

*Line 602 "regionalization": What is this? Do you mean optimization (or tuning) of hydrological parameters to reproduce historical records?*

"Regionalization" is a term mentioned by Samaniego et al. (2017) themselves. It means creating spatially variable parameter fields at the required resolution. We have added "(creating spatially variable parameter fields)" to explain in the text (see lines 597-600).

*Line 603 "scale-consistent flux-preserving": Hard to know what is meant here. Rephrase.*

Rephrased to "parameterizations that yield the same hydrological fluxes at different resolutions." (see lines 597-600).

*Line 604 "parameterization is possible": What kind of parameterization is mentioned here? What does possible mean?*

This part has been removed due to the change mentioned under 603.

*Line 623 Table 1: It is interesting to compare with earlier works (e.g. Table 2 and 5 of Hanasaki et al. 2018).*

We have added the results of Hanasaki et al. (2018) to Table 2.

*Line 635 "cross-correlation": Which did you use cross-correlation (a technique frequently used in signal processing) or Pearson's correlation? I'm asking this because the results indicate the authors used Pearson's correlation, but they always wrote cross-correlation throughout the text.*

Indeed, we used Pearson's correlation coefficient. We used cross-correlation because we are comparing two time series (measured and simulated) as opposed to autocorrelation. Several of the authors have done time series modelling (Box and Jenkins modelling) before, hence the use of this term. However, as it seems confusing we have removed "cross" and just use "correlation" in the revised manuscript.

*Line 647 "Figure 3": It is hard to see the differences between a (30min) and b (5min). Why don't you show the difference between two?*

This is a valid point. We now show the correlation coefficient of 5 arc-minute in panel (a) and the difference between 5 arc-minute and 30 arc-minute in panel (b).  Please check Figure 2 in the revised manuscript).

*Line 648 "Figure 4": As Figure 3 clearly indicates, the selected GRDC stations are concentrated in Europe. Figure 4 might be a bit misleading if the majority of stations are concentrated in some specific regions (e.g. snow dominated stations in Europe). My suggestion is to make Figure 4 for major climatic zones. It would be useful to identify in which climatic conditions the results are improved. As mentioned above, it would be also interesting to separate Figure 4 by catchment area. I suspect the improvements are concentrated in relatively small basins. Another point is that total frequency apparently far exceeds the number of stations (3597). Elaborate how to see these panels (same for Figure 5).*

Starting with the total frequency:  we thank the reviewer for noting this plotting error which we now corrected. Also, as a result of this error we wrongly plotted the different catchment sizes in the histograms, which we also corrected.

Upon the suggestion of the reviewer we have calculated the measures for different climate zones. We have classified the stations to the Köppen-Geiger climate zones A (Tropical), B (Desert), C (Temperate) and D (Continental). We have excluded the arctic climates as we have hardly any GRDC stations for these regions. We have calculated for each zone the cumulative distribution of the KGE values and plotted the result from the 5 and 30 arc-minutes simulation in one figure for easy comparison. As can be seen from the figures below, the improvement is equally visible climate zones A, B and C  and less so for D.   Climate zone D is somewhat under-represented in the dataset due to the low densities over Russia, but well represented in the U.S. So, it may indeed be that the improvement is somewhat biased because of the under-representation of the continental zone in the GRDC dataset. We have added a sentence to this effect.

[Figure]

[Figure]

We also checked if the effect of altitude is important in explaining the differences between the 5 and 30 arc-minutes results. The Figure below shows that this is indeed important. The improvements are notable better for GRDC stations at higher altitude than at lower altitude.

[Figure]

So the resolution has an effect on catchments that are positioned higher because together with the temperature lapse rate, the snow dynamics are better captured at higher resolution.

Finally, in the new plots (see the revised paper) it is now evident that indeed we see a shift to higher KGE and correlation coefficients for the smaller catchments in particular.

So, based on these analyses we changed the paper as follows. In order to limit the paper size we decided not to include the relation between KGE and climate zones, but shortly mention the results of this analysis. We have however included the figure comparing the KGE cumulative distribution plots for GRDC stations below and above 1000 meter and have added comments about the impact of catchment size.

Apart from a new figure (Figure 4 in the revised manuscript), the explanation for the differences between 5 and 30 arc-minutes now reads (see lines 650-667).

"It is difficult to exactly assess which of these factors are most important in determining the improvement. Inspecting the histograms of correlation and KGE (Figure 3) shows that the improvement is mostly apparent for the smaller sized catchments, which supports the notion that a better delineation of the catchments' shape, topography and drainage network could be the cause. However, disentangling these individual effects would require further study. To investigate the possible effects of better snow dynamics we classified the GRDC stations into stations below 1000 m altitude (above mean sea-level) and those above 1000 m.  The GRDC stations above 1000 m are expected to experience precipitation falling as snow during periods of the year. The Results in Figure 5 clearly show that the improvement is larger for the higher GRDC stations, This supports the explanation that better snow dynamics due to temperature lapsing in combination with a better resolved digital elevation model is partly responsible for the better results at 5 arc-minutes. We also investigated if improvements were notably different between climate zones, by separately calculating KGEs for GRDC stations in the Köppen-Geiger zones A (Tropical), B (Desert), C (Temperate) and D (Continental). The results (not shown) show that the improvement is equally visible for climate zones A, B and C  and less so for D (continental). Without further analysis this is difficult to explain. Note however that the continental climate zone is somewhat under-represented in the GRDC dataset due to the low densities over Russia, although it is well represented in the U.S. So, it may be that the global improvements shown in Figure 3 are somewhat positively biased."

*Line 694 "in case of the Niger River, not representing the inner delta. . .": Or simply something wrong with input or validation data.*

This could indeed be the case, but there is no strong indication for this. Looking for an explanation, we would rather look at the model itself.  We know that floodplain inundation and evaporation is important in the Niger inner Delta and that we have not accounted for it in the simulation. So, this would be the prime candidate for an explanation of the lack of improvement.

*Line 653 "a better delineation of the outline of the basins. . .": You mentioned that the error in catchment area is less than 15% for all basins for either spatial resolutions. Further elaborate what do you mean by "better delineation of the outline" here.*

We mean a delineation of the "shape" of the catchment. The size could be well represented, but still there could be errors in the shape. We have changed "outline" to "shape" in the revised manuscript (see line 644-654).

*Line 656 "better snow dynamics due to the downscaling of temperature to 5 arc-minute resolution":*
*Similar comment to above. This argument must be easily supported by showing the performance of snow*
*dominated regions for two simulations (i.e. excluding snow free regions from Figure 4).*

This argument is now supported by an extra figure (see remarks above).

*Line 670 "Although results are generally better, the spatial distribution of results is similar to those found*
*by Van Beek et al. (2011) for PCR-GLOBWB1". This conveys hardly any information. What does "generally*
*better" mean? What are similar and what are not?*

We agree that this does not add much and removed the sentence.

*Line 688 "indicating a higher skill with regard to capturing extremes and anomalies": I'm not convinced*
*at all. As mentioned above, the performance must be different by catchment area, climate, topography*
*and other factors. Show concrete evidences for this claim.*

We have already shown the relationships between performance and catchment area, climate and
topography. Here we deal with the anomaly correlation, which is the correlation after the mean
seasonal variation has been removed. Out of necessity this explains the ability of the model to represent
extremes (within-year differences from seasonal averages) and (inter-annual) anomalies. The Figure
shows that results are better for the 5 arc-minute simulation. So, the statement as such is correct. We
agree that it is difficult then to further explain why anomalies are better captured by the high-resolution
model.

*Line 763 "Also, Figure 10 shows that agricultural water withdrawal is underestimated . . .": I'm quite*
*puzzled by the right panel of Figure 10a. First, majority of plots are located far below the y=x line*
*indicating most countries are underestimated one (or two) order of magnitude. Next, although majority*
*of countries are strongly underestimated, the correlation slope is larger than 1 which indicates the*
*overall results are overestimated due to some outliers' behavior. These points should be more*
*highlighted to call readers' attention. Finally, my honest interpretation of Figures 9 and 10 is that this*
*model fails to reproduce the historical dynamics of country-specific water withdrawal. At best, the*
*simulation outputs are considerably different from AQUASTAT. Further clarify the authors' intention to*
*show Figures 9 and 10 together with discussion on the capability and limitation of the water use module*
*of PCR-GLOBWB 2.*

We agree, as stated in the beginning of this rebuttal, that we could have been a bit more critical when
discussing the validation results of the water withdrawal data. We will do so subsequently and also in
the manuscript. We respectfully disagree with the statement that the model "fails to reproduce the
historical dynamics" which is much too strong. We capture the most important water users by source and also for irrigation water use, including the increase thereof over the years. Also, we really should mention that none of the previous references shown in e.g. Table 2 have compared water withdrawal to AQUASTAT data *per year, per country per sector and per water source*. Per sector only total water withdrawal has been compared and per country the source of water (Wada et al., 2014). Nevertheless, we should say that:

1) We underestimate groundwater withdrawal for the smaller water users, which can be explained by groundwater use by farmers in summer time for countries for which areas are not registered as irrigation in MIRCA, e.g. Germany and the Netherlands, but which are reported in AQUASTAT.

2) We underestimate irrigation water use for the smaller water users. This is related to the fact that in many of the smaller water use countries, water is used for irrigation only occasionally in dry summers. Thus these areas are not mapped as irrigated crops in MIRCA, or they use irrigation technology that is not part of MIRCA, e.g. subsurface drainage by artificially high surface water levels such as in a number developed delta regions in the world. The fact that these smaller countries are not well represented still means that we are able to capture the big water users, which are most important for global scale analyses. The fact that the slopes are still close to one in Figure 10 comes from the fact that the regression lines have been fit at the original scale and the resulting high leverage of the big users (see the Figure below). We have kept it this way, because we do want to stress the importance to be able to simulate the large quantities of water withdrawal that impact the hydrological cycle significantly.

[Figure]

3) The underestimation of industrial water withdrawal is caused by the fact that we do not include water withdrawal for thermo-electric cooling of power plants.

4) The underestimation of domestic water withdrawal comes from the fact that we assume that the priority of water allocation is proportional to demand. This means that in times of shortage, water withdrawal is reduced with and equal percentage for agriculture, industry and domestic. In many countries however, there is a priority series, whereby domestic demand is first met, industrial demand next and agricultural demand comes last. As a result, we underestimate withdrawal. This is also partly the cause for the underestimation of industrial water withdrawal. This is corroborated by plotting gross water demand (which would be withdrawal if no shortage would occur) against Aquastat data that show a much better fit with regression coefficients closer to 1 for domestic water demand (see the figures below). This shows that improvements are in order in our water allocation scheme.

Accordingly, the description of Figures 9 and 10 now read as follows (see lines 782-813).

"We compared simulated water withdrawal data from PCR-GLOBWB 2 with reported withdrawal data per country from AQUASTAT (FAO, 2016). The results are shown subdivided per source (Figure 10) and per sector (Figure 11). Total water withdrawal and surface water withdrawal are simulated reasonably well ($R^2$ between 0.84 and 0.96 and regression slopes between 0.70 and 1.08). However, groundwater withdrawal is underestimated for the smaller water users.  A likely explanation for this is occasional groundwater withdrawal by farmers during dry periods in areas that have not been mapped as irrigated crops in MIRCA, such as grasslands in e.g. Germany and the Netherlands, while this groundwater withdrawal is reported in AQUASTAT.

When looking at water withdrawal per sector, results are mixed. The largest agricultural water users are well captured, but the smaller ones are clearly underestimated. This is related to the fact that in many regions of the smaller water use countries, water is used for irrigation only occasionally during dry summers, while these areas are not mapped as irrigated crops in MIRCA. Also, many of these countries use irrigation technology that is not part of MIRCA, e.g. subsurface drainage by artificially high surface water levels such as in a number developed delta regions in the world. However, even though these smaller countries are not well represented, PCR-GLOBWB 2 is still able to capture the big water users, which have a significant impact on the water cycle and are most important for global scale analyses.

Both industrial and domestic water withdrawals are underestimated. The underestimation of industrial water withdrawal is partly caused by the fact that we do not include water withdrawal for thermo-electric cooling of power plants. The underestimation of domestic water withdrawal comes from the fact that we assume that the priority of water allocation is proportional to demand. This means that in times of shortage, water withdrawal is reduced with an equal percentage for agriculture, industry and domestic use. In many countries however, there is a priority series, whereby domestic demand is first met, industrial demand next and agricultural demand comes last. As a result, we underestimate domestic water withdrawal and it also partly causes the underestimation of industrial water withdrawal. This is corroborated by plotting gross water demand (which would be withdrawal if no shortage would occur) against AQUASTAT data. These plots (not shown here) result in the regression slopes of 0.68-0.75 for industrial demand and 0.78-0.92 for domestic demand. These results thus reveal that the water allocation scheme of PCR-GLOBWB 2 should be further improved."

[Figure]

a) Country PCR-GLOBWB industrial demand and AQUASTAT industrial withdrawal (km3/year) - log scale b) Country PCR-GLOBWB domestic demand and AQUASTAT industrial withdrawal (km3/year) - log scale

*Line 786 "Simulated water withdrawal, by source and sector, matches reasonably well with reported water withdrawal from AQUASTAT": I'm not able to agree with this statement.The authors reported that the regression slope was as low as 0.54 for some cases (line 761).*

Agreed. See the previous points.

*Technical comments*

*Line 67 "Schewe et al. 2013; Haddeland et al. 2013": Check publication year of these articles. It must be 2014.*

Changed accordingly

*Line 89 "collectively over 2100 references": do you mean citations?*

Yes. We have changed this accordingly.

**References:**

[revised manuscript text omitted]